

# The Finite-volumE Sea ice–Ocean Model (FESOM2)

Sergey Danilov[1,2], Dmitry Sidorenko[1], Qiang Wang[1], and Thomas Jung[1]

[1]Alfred Wegener Institute for Polar and Marine Research, Bremerhaven, Germany
[2]A. M. Obukhov Institute of Atmospheric Physics RAS, Moscow, Russia

*Correspondence to:* S. Danilov (sergey.danilov@awi.de)

**Abstract.** Version 2 of the unstructured-mesh sea ice – ocean circulation model FESOM is presented. It builds upon FESOM1.4 (Wang et al., 2014, Geosci. Mod. Dev., 7, 663–693) but differs by its dynamical core (finite volumes instead of finite elements) and is formulated using the Arbitrary Lagrangian Eulerian (ALE) vertical coordinate, which increases model flexibility. The model inher-

its the framework and sea ice model from the previous version, which minimizes the efforts needed from a user to switch from one version to the other. The ocean states simulated with FESOM1.4 and FESOM2.0 driven by CORE-II forcing are compared on a mesh used for CORE-II intercomparison project. Additionally the performance on an eddy-permitting mesh with uniform resolution is discussed. The new version improves numerical efficiency of FESOM in terms of CPU time by at

least three times while retaining its fidelity in simulating sea ice and ocean. From this it is argued that FESOM2.0 provides a major step forward in establishing unstructured-mesh models as valuable tools in climate research.

## 1 Introduction

Ocean circulation models formulated on unstructured meshes offer multi-resolution functionality in

a seamless way. Although they are common in coastal ocean modelling, they are only beginning to be used for global ocean studies. The Finite-Element Sea-ice Ocean circulation Model (FESOM, Wang et al. (2014)) is the first mature global multi-resolution model designed to simulate the large-scale ocean. A number of FESOM-based studies related to the impact of local dynamics on the global ocean (see, e.g., Hellmer et al. (2012), Haid and Timmermann (2013), Wekerle et al. (2013), Haid

et al. (2015), Wang et al. (2016a), Sein et al. (2016), Wekerle et al. (2016)) indicate that the multi-resolution approach advocated by FESOM is successful and allows one to explore the impact of local processes on the global ocean with moderate computational effort (see Sein et al. (2016)). Other new





global multi-resolution models are appearing (see Ringler et al. (2013) and Korn (2016)), and new knowledge on unstructured-mesh modeling has accumulated (for review see Danilov (2013)). Al-
though FESOM1.4 (Wang et al. (2014)) already offers a very competitive throughput compared to structured-mesh models in massively parallel applications (Sein et al. (2016)), we continue to explore the ways to further increase the numerical efficiency of unstructured-mesh models and extend their area of applicability. This manuscript describes the new numerical core of FESOM2 which is based on finite-volume discretization. Despite the change of the discretisation type, we keep the old
abbreviation, which now will take 'E' from the last letter of 'volume'. The reason is that FESOM2 builds on the framework of FESOM1.4, including its ice component, FESIM (Danilov et al. (2015)), its input and output routines and its user interface. It works on general triangular meshes and is conceived so that no new learning is required for users having experience with FESOM1.4. We will use FESOM2 as a root name for the new version, and FESOM2.0 for the implementation available at
present.

The main reason for switching to a new finite-volume numerical core in FESOM2 is its higher computational efficiency. It stems largely from a more efficient data structure. FESOM1.4 is based on tetrahedral elements, and tetrahedra below any surface triangle do not necessarily keep the same neighbourhood connectivity pattern as the depth increases. 3D auxiliary and look-up arrays are there-
fore needed, and accessing them for each element slows down the performance. Another reason for switching to a finite volume version is the availability of clearly defined fluxes and a possibility to choose from a selection of transport algorithms, which was very limited for the continuous Galerkin discretization of FESOM1.4. A very useful feature of FESOM1.4 is its ability to combine geopotential and terrain following vertical mesh levels. Namely it was the reason for using tetrahedral
elements and not triangular prisms. To ensure similar functionality in the new version, we introduce the Arbitrary Lagrangian Eulerian (ALE) vertical coordinate (see, e.g., Donea and Huerta (2003)), which provides a general approach to incorporate different types of vertical coordinate within the same code.

Although many details of the finite-volume method used by FESOM2 have already been presented
in Danilov (2012), we will repeat their description here for completeness. Besides, the ALE vertical coordinate redefines the implementation details. The paper begins with the description of basic model numerics, delegating some details and implementation variants to Appendices. The performance of FESOM2 is compared to that of FESOM1.4 in simulations driven by the CORE-II forcing (Large and Yeager (2009)). We report on simulations carried out on a coarse (nominally 1°) mesh
used by FESOM1.4 in the framework of CORE-II intercomparison, and on a global mesh with a resolution about 15 km. The intention here is to illustrate that FESOM2 is a fully functional and highly competitive general ocean circulation model. A detailed model assessment paper will be presented separately.



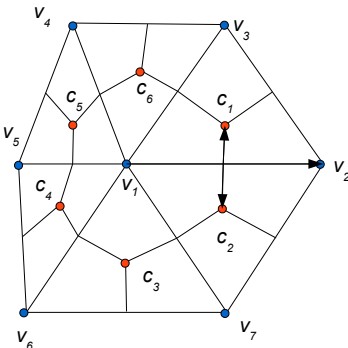

Figure 1: Schematic of cell-vertex discretization. The horizontal velocities are located at cell (triangle) centers (red circles) and scalar quantities (the elevation, pressure, temperature and salinity) are at vertices (blue circles). The vertical velocity and the curl of horizontal velocity (the relative vorticity) are at the scalar locations too. Scalar control volumes (here the volume associated to vertex $v_1$ is shown) are obtained by connecting the cell centers with midpoints of edges. An edge is defined as the ordered pair of connected vertices. Edge $e$ connecting vertices $v_1$ and $v_2$ is characterized by the list of its vertices $v(e) = (v_1, v_2)$ and the ordered list of cells $c(e) = (c_1, c_2)$ with $c_1$ on the left. The edge vector $\mathbf{l}_e$ connects vertex $v_1$ to vertex $v_2$. The edge cross-vectors $\mathbf{d}_{ec_1}$ and $\mathbf{d}_{ec_2}$ connect the edge midpoint to the respective cell centers. Each cell is characterized by the list of its vertices $v(c)$ which is $(v_1, v_2, v_3)$ for $c = c_1$ and the list of its nearest neighbors $n(c)$. For $c = c_1$, $n(c)$ includes $c_2$, $c_6$ and a triangle (not shown) across the edge formed by $v_2$ and $v_3$. One can also introduce $c(v)$ which is $(c_1, c_2, c_3, c_4, c_5, c_6)$ for $v = v_1$, and other possible lists.

## 2 Basic description

### 2.1 The placement of variables

FESOM2 uses a cell–vertex placement of variables in the horizontal directions. The 3D mesh structure is defined by the surface triangular mesh and a system of level surfaces which form a system of prisms. In a horizontal plane, the horizontal velocities are located at cell (triangle) centroids, and scalar variables are at mesh (triangle) vertices. The vector control volumes are the prisms based on mesh surface cells, and the prisms based on median-dual control volumes are used for scalars (temperature, salinity, pressure and elevation). The latter are obtained by connecting cell centroids with edge midpoints, as illustrated in Fig. 1. The same cell–vertex placement of variables is also used in FVCOM (Chen et al. (2003)).





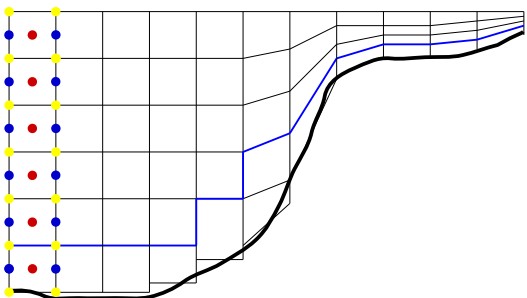

Figure 2: Schematic of vertical discretization. The thick line represents the bottom, the thin lines represent the layer boundaries and vertical faces of prisms. The location of variables is shown for the left column only. The blue circles correspond to scalar quantities (temperature, salinity, pressure), the red circles to the horizontal velocities and the yellow ones to the vertical exchange velocities. The bottom can be represented with full cells (three left columns) or partial cells (the next two). The mesh levels can also be terrain following, and the number of layers may vary (the right part of the schematic). The layer thickness in the ALE procedure may vary in prisms above the blue line. The height of prisms in contact the bottom is fixed.

In the vertical direction, the horizontal velocities and scalars are located at mid-levels. The velocities of inter-layer exchange (vertical velocities for flat layer surfaces) are located at full layers and at scalar points. Figure 2 illustrates this arrangement.

The layer thicknesses are defined at scalar locations (to be consistent with the elevation). There are also auxiliary layer thicknesses at the horizontal velocity locations. They are interpolated from the vertex layer thicknesses.

The cell-vertex discretization selected for FESOM2 can be viewed as an analog of an Arakawa B-grid (see also below) while that of FESOM1.4 is an analog of A-grid. The cell-vertex discretization is free of pressure modes, which would be excited in the A-grid FESOM1.4 without its stabilization. However, the cell-vertex discretization allows spurious inertial modes because of excessively many degrees of freedom used to represent the horizontal velocities. They can be filtered by the horizontal viscosity. In the quasi-hexagonal C-grid discretization used by MPAS (Ringler et al. (2013)) the location of scalar variables is the same (on vertices of dual triangular mesh) as in FESOM2. The triangular C-grid of ICON (www.mpimet.mpg.de/en/ science/models/icon/) is notably different for its scalar variables are located at cells and there are twice as many of them as in FESOM2. Our preference to the cell–vertex discretization is mostly due to its lack of pressure modes, the straightforward way of handling its spurious modes and the ability to work on general triangular meshes (in



contrast to orthogonal meshes required by C-grids). Such meshes are more flexible than the Voronoi quasi-hexagonal meshes or orthogonal triangular meshes needed for C-grids.

## 2.2 Notation

For convenience of model description we introduce the following notation. Quantities defined at cell centroids will be denoted with the lower index $c$, and the quantities at vertices will be denoted with the lower index $v$. The vertical index $k$ will appear as the first index, but it will be suppressed if this does not lead to ambiguities. The agreement is that the layer index increases downwards. The indices may appear in pairs or in triples. Thus the pair $kc$ means the vertical layer (or level for some quantities) $k$ and cell $c$, and the triple $kcv$ means that the quantity relates to layer (level) $k$, cell $c$ and vertex $v$ of this cell. We use the notation $c(v)$ for the list of cells that contain vertex $v$, $v(c)$ for the list of vertices of cell $c$, $e(v)$ for the list of edges emanating from vertex $v$ and so on. Each edge $e$ is characterized by its vertices $v(e)$, the neighboring cells $c(e)$, the length vector $\mathbf{l}_e$ directed from the first vertex in $v(e)$ to the second one and two cross-edge vectors $\mathbf{d}_{ec}$ directed from the edge center to the cell center of the left and right cells respectively (see Fig. 1). The cells in the list $c(e)$ are ordered so that the first one is on the left of the vector $\mathbf{l}_e$. The boundary edges have only one (left) cell in the list $c(e)$.

We use spherical coordinate system with the north pole displaced to Greenland (commonly 75°N, 50°W). The metrics is taken cellwise-constant. The vectors $\mathbf{d}$ are stored in local physical measure for they always enter in combination with velocity to give normal transports. Vectors $\mathbf{l}$ are stored in radian measure. We will skip details of spherical geometry and ignore the difference in the representation of $\mathbf{l}$ and $\mathbf{d}$ for brevity below. The $x$ and $y$ directions should be understood as local zonal and meridional directions.

## 2.3 Bottom representation

The bottom topography is commonly specified at scalar points because the elevation is defined there. However, for discretizations operating with full velocity vectors, this would imply that velocity points are also at topographic boundaries. In this case the only safe option is to use the no-slip boundary conditions, similar to the traditional B-grids. To avoid this constraint, we use the cellwise representation of bottom topography. In this case both no-slip and free slip boundary conditions are possible. Their implementation relies on the concept of ghost cells which are obtained from the boundary elements by reflection with respect to the boundary face (edge in 2D). The drawback of the elementwise bottom representation is that the total thickness is undefined at scalar points if the bottom is stepwise (geopotential vertical coordinate). The motion of level surfaces of the ALE vertical coordinate at each scalar location is then limited to the layers that do not contact the bottom topography (above the blue line in Fig. 2). This is related to the implementation of partial cells which is much simpler if the thickness of the bottom layer stays fixed. The layer thickness $h$ is dynami-





cally updated at scalar points (vertices) in the layers that are affected by the ALE algorithm and interpolated to the cells

$$h_c = (1/3) \sum_{v(c)} h_v.$$

The cell thicknesses $h_c$ enter the discretized equations as the products with horizontal velocities.

Because of cellwise bottom representation, algorithms aiming to closely follow the bottom topography may create triangular prisms pointing into land (two lateral faces touch the land) at certain levels on $z$-coordinate meshes even if such prisms were absent along the coast. Such prisms lead to instabilities in practice and have to be excluded. The opposite situation with land prisms pointing into the ocean is much less dangerous, yet it is better to avoid them too. We adjust the number of

layers under each surface triangle at the stage of mesh design to exclude such potentially dangerous situations. This issue is absent in FESOM1.4 because of the difference in the placement of horizontal velocities and no-slip boundary conditions. Since the number of cells is nearly twice as large as the number of vertices, the cellwise bottom representation may contain more detail than can be resolved by the field of vertical velocity. This may make quasi-vertical transport velocities looking noisy in

layers adjacent to the bottom.

### 2.4    Partial cells

Partial cells on $z$-coordinate meshes are naturally taken into account in the ALE formulation because it always deals with variable layer thicknesses (heights of prisms). If $K_c$ is the number of layers under cell $c$, we define

$$K_v^+ = \max_{c(v)} K_c, \quad K_v^- = \min_{c(v)} K_c.$$

If the layer thickness are varied in the ALE procedure, this is limited to $K_v^- - 1$ layers. With this agreement, the thickness of the lowest layer on cells is kept as initially prescribed. In this case the implementation of partial cells reduces to taking the thicknesses of the lowest layers on cells as

dictated by the bottom topography unless they are too thick (the real depth is deeper than the deepest standard level by more than half thickness of the last standard layer), in which case we bound them. The heights of scalar control prisms in the layers below $K_v^-$ are formally undefined, so they are considered to be the volume-mean ones. Scalar and vector quantities defined at mid-layers are kept at their standard locations. This avoids creating spurious pressure gradients. The partial cells then

work through the modified transports crossing the faces of control volumes. Since the horizontal velocities are located at cells, the transports entering scalar control volumes are uniquely defined. For vector control volumes the areas of vertical faces may be different on two prisms meeting through the face. Taking the minimum area to compute fluxes is the safest option in this case.





### 3 Layer equations and time stepping

#### 3.1 Layer thicknesses and layer equations

We introduce layer thicknesses $h_k = h_k(x, y, t)$, where $k = 1 : K$ is the layer index and $K$ the total number of layers. They are functions of the horizontal coordinates and time in a general case. We basically follow the implementation of ALE vertical coordinate as presented in Ringler et al. (2013) (there are other approaches, see, e.g., Adcroft and Hallberg (2006), Hofmeister et al. (2010)). Namely, we introduce the transport velocities $w$ through the top and bottom boundaries of the prisms. They are the differences between the physical velocities in the direction normal to the layer interfaces and the velocities due to the motion of the interfaces. These velocities are defined at the interfaces (the yellow points in Fig. 2). For layer $k$ the top interface has index $k$ and the bottom one is $k + 1$. All other quantities — horizontal velocities $\mathbf{u}$, temperature $T$, salinity $S$ and pressure $p$ are defined at mid-layers. Their depths will be denoted as $Z_k$, and the notation $z_k$ is kept for the depths of mesh levels (the layer interfaces). They are functions of horizontal coordinates and time in a general case.

The equations of motion, continuity and tracer balance are integrated vertically over the layers. We will use $T$ to denote an arbitrary tracer. The continuity equation becomes the equation on layer thicknesses

$$\partial_t h + \nabla \cdot (\mathbf{u}h) + (w^t - w^b) + W\delta_{k1} = 0, \tag{1}$$

and the tracer equation becomes

$$\partial_t (hT) + \nabla \cdot (\mathbf{u}hT) + (w^t T^t - w^b T^b) + W T_W \delta_{k1} = \nabla_3 \cdot h\mathbf{K}\nabla_3 T. \tag{2}$$

Here, $W$ is the water flux leaving the ocean at the surface, it contributes to the first layer only (hence the delta-function); $T_W$ is the property transported with the surface water flux and the indices $t$ and $b$ imply the top and the bottom of the layer. The operator $\nabla$ is a two-dimensional one. The right hand side of (2) contains the 3 by 3 diffusivity tensor $\mathbf{K}$, and $\nabla_3$ denotes the 3D divergence or gradient operators. In writing the 3D divergence we assume the discrete form $\nabla(...) + ((...)^t - (...)^b)/h$, where $(...)$ are the placeholders for the horizontal and vertical components of 3D vector it acts on. The components of 3D gradient do not share the same location, so the discretization of $\mathbf{K}\nabla_3 T$ requires special care (see Lemarié et al. (2012a) for the discussion for quadrilateral meshes). Note that $w$ coincides with the vertical velocity through the layer surface only if the layer surfaces are flat. If the surfaces are inclined, $w$ is the quasi-vertical transport velocity defining the exchange between the layers.

Integrating (1) vertically and assuming $w^t = 0$ at the free surface, we obtain the elevation equation

$$\partial_t \eta + \nabla \cdot \sum_k h_k \mathbf{u}_k + W = 0. \tag{3}$$





The layer-intergated momentum equation in the flux form is

$$\partial_t(h\mathbf{u}) + \nabla \cdot (h\mathbf{u}\mathbf{u}) + w^t\mathbf{u}^t - w^b\mathbf{u}^b + f\mathbf{k} \times \mathbf{u}h + h(\nabla p + g\rho\nabla Z)/\rho_0 =$$
$$D_{uh}\mathbf{u} + (\nu_v\partial_z\mathbf{u})^t - (\nu_v\partial_z\mathbf{u})^b, \tag{4}$$

with $D_{uh}\mathbf{u}$ the horizontal viscosity operator (to be specified later), $\nu_v$ the vertical viscosity coefficient, $f$ the Coriolis parameter and $\mathbf{k}$ a unit vertical vector. We ignore the momentum source due to the added water $W$ at the surface. The pressure field is expressed as

$$p = p_a + g\rho_0\eta + p_h, \quad p_h = g\int_z^0 \rho dz.$$

with $p_a$ the atmospheric pressure, which will be omitted for brevity, $\eta$ the elevation, $\rho$ the deviation of density from its reference value $\rho_0$, and $p_h$ the hydrostatic pressure due to $\rho$. The term with the pressure gradient, $g\rho\nabla Z$, accounts for the fact that layers deviate from geopotential surfaces. The quantity $Z$ appearing in this term is the $z$-coordinate of the midplane of the layer with the thickness $h$. The origin of this term should become clear if one recalls that the horizontal pressure gradient has to be computed at constant vertical coordinate $z$.

If the flux form (4) is used, it is more natural to formulate the solution procedure in terms of the horizontal layer transport velocities $\mathbf{U} = h\mathbf{u}$.

We get another familiar option by subtracting $\mathbf{u}$ multiplied with the thickness equations (1), rearranging the terms with vertical transports and dividing over the layer thickness $h$:

$$\partial_t\mathbf{u} + \frac{\omega + f}{h}\mathbf{k} \times \mathbf{u}h + ((w\partial_z\mathbf{u})^t + (w\partial_z\mathbf{u})^b)/2 + \nabla(p/\rho_0 + \mathbf{u}^2/2) + g\rho\nabla Z/\rho_0 =$$
$$D_u\mathbf{u} + \partial_z(\nu_v\partial_z\mathbf{u}). \tag{5}$$

Here, additionally, we used the identity

$$\mathbf{u} \cdot \nabla\mathbf{u} = \omega\mathbf{k} \times \mathbf{u} + \nabla(\mathbf{u}^2/2), \quad \omega = \mathbf{k} \cdot (\nabla \times \mathbf{u}),$$

which leads to the vector-invariant form of the momentum equation.

The second term on the lhs of (3.1) includes division and multiplication with the layer thickness, and in doing so, it introduces the layer potential vorticity (PV), $q = (\omega + f)/h$ and its transport $\mathbf{u}h$. The layer thickness drops out from the continuous equation (3.1). In the discrete case, the location of vorticity points (vertices) and velocity points is different, and by keeping $h$ the equation will then operate on the same horizontal transports as the thickness equations. This is the prerequisite for developing discretizations that conserve potential vorticity. We will suppress $h$ in () for simplicity further, but including it requires only small modifications.

To summarize, the velocity $w$ of quasi-vertical transport through the interfaces replaces the vertical velocity in the formulas above. The layer surfaces can be any combination of the standard choices, including the moving surfaces.



### 3.2 Asynchronous time stepping

FESOM1.4 uses asynchronous time stepping, with the horizontal velocities and scalars shifted by a half time step. We adapt it to FESOM2. This requires that the elevation and layer thicknesses be introduced at respectively full (integer) and half-integer time levels. We write

$$h^{n+1/2} - h^{n-1/2} = -\tau[\nabla \cdot (\mathbf{u}^n h^*) + w^t - w^b + W^{n-1/2}\delta_{k1}]$$

and

$$h^{n+1/2}T^{n+1/2} - h^{n-1/2}T^{n-1/2} = -\tau[\nabla \cdot (\mathbf{u}^n h^* T^n) + w^t T^t - w^b T^b + W^{n-1/2}T_W\delta_{k1}] + D_T,$$

to warrant tracer conservation. Here $\tau$ is the time step and $D_T$ stands for the terms related to diffusion. We omit time index on $w$, for $w$ is related to $\mathbf{u}$ and $h$. Since the horizontal velocity is centered in time, these equations will be of the second order for advective terms if $h^* = h^n$. When the vector-invariant form of momentum equation is used, taking $h^* = h^{n-1/2}$ is more convenient. In this case one does not need thicknesses at full time levels, but only the elevation. Although this formally reduces the time order to the first, the consequences are minor as long as thickness variations are small, which are our options at present. Besides, the elevation is usually computed with the accuracy shifted to the first-order in large-scale ocean models, including this one. We will proceed with this option here. Appendix A shows how to implement $h^* = h^n$ for the flux form of momentum equation and its generalizations are straightforward.

The elevation at full time steps and the total thickness on half-steps, given by the vertical sum of $h_k$, may become decoupled due to numerical errors. In order to suppress such decoupling we seek for an algorithm which maintains consistency between the physical layer thickness ($h$, used with tracers) and dynamical thickness (dependent on the elevation $\eta$). We introduce

$$\overline{h} = \sum_k h_k - H,$$

where $H$ is the unperturbed ocean thickness. $\overline{h}$ would be identical to the elevation $\eta$ in the continuous world, but not in the discrete formulation here.

For $h^* = h^{n-1/2}$ we write for the elevation

$$\eta^{n+1} - \eta^n = -\tau(\alpha(\nabla \cdot \int^{\overline{h}^{n+1/2}} \mathbf{u}^{n+1}dz + W^{n+1/2}) + (1-\alpha)(\nabla \cdot \int^{\overline{h}^{n-1/2}} \mathbf{u}^n dz + W^{n-1/2})).$$

Here $\alpha$ is the implicitness parameter ($0.5 \leq \alpha \leq 1$) in the continuity equation. Note that the velocities at different time steps are taken on their respective meshes. This approach is inspired by Campin et al. (2004). The equation for thicknesses can be vertically integrated giving, under the condition that the surface value of $w^t$ vanishes,

$$\overline{h}^{n+1/2} - \overline{h}^{n-1/2} = -\tau\nabla \cdot \int^{\overline{h}^{n-1/2}} \mathbf{u}^n dz - \tau W^{n-1/2}. \tag{6}$$





Expressing the rhs in the formula for $\eta$ through the difference in surface displacements $\overline{h}$ from the last formula we see that $\eta$ and $\overline{h}$ can be made consistent if we require

$$\eta^n = \alpha\overline{h}^{n+1/2} + (1-\alpha)\overline{h}^{n-1/2}. \tag{7}$$

Now, to eliminate the possibility for $\eta$ and $\overline{h}$ to diverge we always compute $\eta^n$ from the last formula, then estimate $\eta^{n+1}$ by solving dynamical equations, and use it only to compute $\mathbf{u}^{n+1}$. On the new time step a 'copy' of $\eta^{n+1}$ will be created from the respective fields $\overline{h}$. We commonly select $\alpha = 1/2$, in this case $\eta^n$ is just the interpolation between the two adjacent values of $\overline{h}$. Note that (7) will be valid for $h^* = h^n$, it is only the upper limits in the integrals above that will be adjusted. The advantage of this approach compared to the synchronous time stepping is that a single version of $w$ centered at full steps is needed. The disadvantage is the additional machinery involving the thicknesses and elevation.

We will continue by providing more detail on the asynchronous time stepping. We write

$$\mathbf{u}^{n+1} - \mathbf{u}^n = \tau(\mathbf{R}_u^{n+1/2} + \partial_z\nu_v\partial_z\mathbf{u}^{n+1} - g\nabla(\theta\eta^{n+1} + (1-\theta)\eta^n)). \tag{8}$$

Here $\theta$ is the implicitness parameter for the elevation, $\mathbf{R}_u^{n+1/2}$ includes all the terms except for vertical viscosity and the contribution from the elevation. We use the second-order Adams–Bashforth (AB) method for the terms related to the momentum advection and Coriolis acceleration, the contribution of pressure $p_h$ does not need the AB interpolation (because it is centered) and the horizontal viscosity is estimated on the level $n$. We write the predictor equation

$$\mathbf{u}^* - \mathbf{u}^n - \tau\partial_z\nu_v\partial_z(\mathbf{u}^* - \mathbf{u}^n) = \tau(\mathbf{R}_u^{n+1/2} + \partial_z\nu_v\partial_z\mathbf{u}^n - g\nabla\eta^n). \tag{9}$$

Solving the three-diagonal operator on the lhs for each column we find the predicted velocity update $\Delta\mathbf{u} = \mathbf{u}^* - \mathbf{u}^n$. [1]

The corrector step is written as

$$\mathbf{u}^{n+1} - \mathbf{u}^* = -g\tau\theta\nabla(\eta^{n+1} - \eta^n). \tag{10}$$

Expressing the new velocity from this equation and substituting the result into the equation for the elevation, we find

$$\frac{1}{\tau}(\eta^{n+1} - \eta^n) - \alpha\theta g\tau\nabla\cdot\int\limits^{\overline{h}^{n+1/2}}\nabla(\eta^{n+1} - \eta^n)dz =$$

$$-\alpha(\nabla\cdot\int\limits^{\overline{h}^{n+1/2}}(\mathbf{u}^n + \Delta\mathbf{u})dz + W^{n+1/2}) - (1-\alpha)(\nabla\cdot\int\limits^{\overline{h}^{n-1/2}}\mathbf{u}^n dz + W^{n-1/2}). \tag{11}$$

Here, the operator part depends on $h^{n+1/2}$, which is the current value of thickness. The last term on the rhs is taken from the thickness computations on the previous time step.

The overal solution strategy is as follows.

---

[1] The vertical viscosity contribution on the rhs can be conveniently added during the assembly of the operator on the lhs.



– Compute $\eta^n$ from (7). Once it is known, compute $\Delta\mathbf{u}$ from (9).

– Solve (11) for $\eta^{n+1}$ and estimate the new horizontal velocity from (10).

– Compute $\overline{h}^{n+3/2}$ from (6).

– Determine layer thicknesses and $w$ according to the options described below.

– Advance the tracers. The implementation of implicit vertical diffusion will be detailed below.

Options for the vertical coordinate:

– Linear free surface: If we keep the layer thicknesses fixed, the time derivative drops out, and the rest gives us the standard equation to compute $w$, starting from the bottom and continuing to the top,

$$w^t - w^b + \nabla \cdot (h\mathbf{u}) = 0.$$

If this option is applied also to the first layer, the freshwater flux cannot be taken into account in the thickness equation. Its contribution to the salinity equation is then through the virtual salinity flux. In this option, $w$ at the (fixed) ocean surface differs from zero, and so do the
tracer fluxes. They do not necessarily integrate to zero over the ocean surface which is why tracer conservation is violated.

– Full (nonlinear) free surface: We adjust the thickness of the upper layer, while the thicknesses of all other layers are kept fixed, $\partial_t h_k = 0$ for $k > 1$. The thickness equations are used to compute $w$ on levels $k = 2 : K_v$ starting from the bottom. The change in the thickness of the
first layer $h_1^{n+3/2} - h_1^{n+1/2}$ is given by (6) written for the respective time interval. In this case there is no transport through the upper moving surface (the transport velocity $w_1$ is identically zero). This option requires minimum adjustment with respect to the standard $z$-coordinate. However, the matrix of the operator in (11) needs to be re-assembled on each time step.

– We can distribute the total change in height $\partial_t \overline{h}$ between several or all eligible layers. Due to our implementation, at *each* scalar horizontal location they can only be the layers that do not touch the bottom topography. If all eligible layers are involved we estimate

$$\partial_t h_k = (h_k^0/\tilde{H})\partial_t \overline{h},$$

where $h_k^0$ are the unperturbed layer thicknesses and $\tilde{H}$ is their sum for all eligible layers. The
thickness of the layers adjacent to the topography is kept fixed. The equation on thickness, written for each layer, is used to compute transport velocities $w$ starting from zero bottom value. This variant gives the so-called $z^*$-coordinate.

– This can be generalized even further. One can use arbitrary distribution of layer thicknesses provided that their tendencies sum to $\partial_t \overline{h}$ over the layers. In particular, requiring that transport





velocities $w$ are zero, isopycnal layers can be introduced. The levels can move with high-pass vertical velocities, leading to the so called $\tilde{z}$ coordinate, see Leclair and Madec (2011); Petersen et al. (2015), or follow density gradients as in Hofmeister et al. (2010). The unperturbed layer thicknesses need not follow the geopotential surfaces and can be terrain following for example. Additional measures may be required in each particular case. For example, for

terrain-following meshes the algorithms of computing pressure gradient should be adjusted to minimize errors in the momentum equation. Updated transport algorithms are also needed (in the spirit of Lemarié et al. (2012b)) to minimize spurious numerical mixing in terrain-following layers. These generalizations are among the topics of ongoing work.

Because of varying layer thicknesses, the implementation of implicit vertical diffusion needs slight adjustment compared to the case of fixed layers. We write, considering time levels $n-1/2$ and $n+1/2$,

$$h^{n+1/2}T^{n+1/2} - h^{n-1/2}T^{n-1/2} = \tau(R_T^n + (K_{33}\partial_z T^{n+1/2})^t - (K_{33}\partial_z T^{n+1/2})^b)$$

and split it into

$$h^{n+1/2}T^* - h^{n-1/2}T^{n-1/2} = \tau R_T^n$$

and

$$h^{n+1/2}(T^{n+1/2} - T^*) = \tau(K_{33}\partial_z(T^{n+1/2} - T^*) + K_{33}\partial_z T^*)|_b^t.$$

Here $R_T$ contains all advection terms and the terms due to the diffusion tensor except for the diagonal

term with $K_{33}$. Note that the preliminary computation of $T^*$ here is a necessity to garantee that a uniform distribution stays uniform (otherwise some significant digits will be lost).

The semi-implicit implementation of the part related to the surface elevation (external mode) implies that an iterative solver must be used to solve the equation on $\eta^{n+1}$. An alternative is the option with subcycling, as detailed in Appendix B.

## 4   Spatial discretization of equations

To obtain the finite-volume discretization, the governing equations are integrated over the control volumes. The flux divergence terms are then, by virtue of Gauss theorem, transformed to the net fluxes leaving the control volumes. All other terms are estimated as mean over the volumes. It is assumed that

$$A_c\mathbf{u}_c = \int_c \mathbf{u}dS,$$

and similarly for the temperature and other scalars,

$$A_{kv}T_{kv} = \int_{kv} TdS.$$



Here $A_c$ and $A_{kv}$ are the horizontal areas of cells and scalar prisms. Note that the scalar areas vary with depth, hence the index $k$ in $A_{kv}$ in the formula above (the index $k$ will be suppressed in some cases). For layer k, $A_{kv}$ is the area of the prism $kv$ including its top face. The area of bottom face is $A_{(k+1)v}$ and may differ from that of the top one if the bottom is encountered. To be consistent in spherical geometry, we use

$$A_{kv} = \sum_{\overline{c}(v)} A_c/3,$$

where $\overline{c}(v)$ is the list of wet prisms containing $v$ in layer $k$.

Since the horizontal velocity is at centroids, its cell-mean value $\mathbf{u}_c$ can be identified with the value of the field $\mathbf{u}$ at the centroid of cell $c$ with the second order of spatial accuracy. For scalar quantities a similar rule is valid only on uniform meshes, but even in this case it is violated in the vicinity of boundaries or topography. This has some implications for the accuracy of transport operators.

### 4.1 Horizontal operators

– **Scalar gradient** takes vertex values of a field and returns the gradient at the cell center:

$$A_c(\nabla p)_c = \int_c \nabla p\, dS = \sum_{e(c)} l_e \mathbf{n}_e \sum_{v(e)} p_v/2,$$

where $\mathbf{n}_e$ is the outer normal to cell $c$. Clearly $l_e \mathbf{n}_e = -\mathbf{k} \times \mathbf{l}_e$ if $c$ is the first (left) cell of $c(e)$. This procedure introduces $\mathbf{G}_{cv} = (G^x_{cv}, G^y_{cv})$ with the $x$- and $y$ component matrices $G^x_{cv}$ and $G^y_{cv}$. They have three non-zero entries for each cell (triangle) which are stored. In contrast to FESOM1.4, where similar arrays are stored for each tetrahedron (and for 4 vertices and 3 directions), here only surface cells are involved.

– **Vector gradient** takes the values of velocity components and returns their derivatives at cell locations. They are computed through the least squares fit based on the velocities on neighboring cells sharing edges with cell $c$. Their list is $n(c)$. The derivatives $(\alpha_x, \alpha_y)$ of the velocity component $u$ are found by minimizing

$$\mathcal{L} = \sum_{n(c)} (u_c - u_n + (\alpha_x, \alpha_y) \cdot \mathbf{r}_{cn})^2 = \min.$$

Here $\mathbf{r}_{cn} = (x_{cn}, y_{cn})$ is the vector connecting the center of $c$ to that of its neighbor $n$. The solution of the minimization problem can be represented as two matrices $g^x_{cn}$ and $g^y_{cn}$, acting on velocity differences $u_n - u_c$ and returning the derivatives. Computations for $v$-component result in the same matrices. The explicit expressions for matrix entries are:

$$g^x_{cn} = (x_{cn}Y^2 - y_{cn}XY)/d,$$
$$g^y_{cn} = (y_{cn}X^2 - x_{cn}XY)/d.$$



Here $d = X^2 Y^2 - (XY)^2$, $X^2 = \sum_{n(c)} x_{cn}^2$, $Y^2 = \sum_{n(c)} y_{cn}^2$ and $XY = \sum_{n(c)} x_{cn} y_{cn}$. The matrices are computed once and stored.

On the cells touching the lateral walls or bottom topography we use ghost cells (mirror reflections with respect to boundary edge). Their velocities are computed either as $\mathbf{u}_n = -\mathbf{u}_c$ or $\mathbf{u}_n = \mathbf{u}_c - 2(\mathbf{u}_c \cdot \mathbf{n}_{nc})\mathbf{n}_{nc}$ for the no-slip or free-slip cases respectively. Here $n$ is the index of the ghost cell, and $\mathbf{n}_{nc}$ is the vector of unit normal to the edge between cells $c$ and $n$. Note that filing ghost cells takes additional time, but allows using matrices $g_{cn}^x$ and $g_{cn}^y$ related to

the surface cells only. Otherwise separate matrices will be needed for each layer. Note also that ghost cells are insufficient to implement the free-slip condition. In addition, the tangent component of viscous stress should be eliminated directly.

We stress that matrices $g_{cn}^x$ and $g_{cn}^y$ return derivatives of velocity components, and not the components of the tensor of velocity derivatives. The latter includes additional metric terms.

– **Flux divergence** takes fluxes nominally defined on cells and returns their divergence on scalar control volumes:

$$A_{kv}(\nabla \cdot \mathbf{F})_v h_v = \sum_{e(v)} \sum_{c(e)} \mathbf{F}_c h_c \cdot \mathbf{n}_{ec} d_{ec},$$

where $\mathbf{n}_{ec}$ is the outer normal to control volume $v$. Clearly, if $v$ is the first vertex in the list $v(e)$, $\mathbf{n}_{ec} d_{ec} = -\mathbf{k} \times \mathbf{d}_{ec}$ if $c$ is the first in the list $c(e)$ (signs are changed accordingly in other cases). While these rules may sound difficult to memorize, in practice computations are done in a cycle over edges, in which case signs are obvious.

In contrast to the scalar gradient operator, the operator of divergence depends on the layer

(because of bottom topography), which is one of the reasons why it is not stored in advance. Besides, the fluxes $\mathbf{F}$ involve estimates of the scalar quantity being transported. Computing these estimates requires a cycle over edges in any case, so there would be no economy even if the matrices of the divergence operator were introduced.

– **Velocity curl** takes velocities at cells and returns the relative vorticity at vertices:

$$A_{kv} \int_v (\nabla \times \mathbf{u}) \cdot \mathbf{k} dS = \sum_{e(v)} \sum_{c(e)} \mathbf{u}_c \cdot \mathbf{t}_{ec} d_{ec},$$

where $\mathbf{t}_{ec}$ is the unit vector along $\mathbf{d}_{ec}$ oriented so as to make an anticlockwise turn around

vertex $v$. If $v$ is the first in the list $v(e)$ and $c$ is the first in the list $c(e)$, $\mathbf{t}_{ec} d_{ec} = \mathbf{d}_{ec}$. This operator also depends on the layer and is not stored.

It can be verified that the operators introduced above are mimetic. For example, the scalar gradient and divergence are negative adjoint of each other in the energy norm and the curl operator applied to the scalar gradient operator gives identically zero. The latter property allows a PV conserving

discretization, but we will not discuss it here.





### 4.2 Momentum advection

FESOM2.0 has three options for momentum advection. Two of them use the flux form and the third one uses the vector invariant form. In spherical geometry the flux form takes an additional term $M\mathbf{k} \times \mathbf{u}$, where $M = u\tan\lambda/r_E$ is the metric frequency, with $\lambda$ the latitude and $r_E$ the Earth radius.

All the options are based on the understanding that the cell-vertex discretization has an excessive number of velocity degrees of freedom on triangular meshes. The implementation of momentum advection must contain certain averaging in order to suppress the appearance of grid-scale noise.

– **Vertex velocity option.** We compute vertex velocities by averaging

$$A_v\mathbf{u}_v h_v = \sum_{\overline{c}(v)} \mathbf{u}_c h_c A_c/3,$$

and use them to compute the divergence of horizontal momentum flux:

$$A_c(\nabla \cdot (h\mathbf{u}\mathbf{u}))_c = \sum_{e(c)} l_e (\sum_{v(e)} \mathbf{n}_e \cdot \mathbf{u}_v h_v)(\sum_{v(e)} \mathbf{u}_v/4).$$

Here $\mathbf{n}_e$ is the external normal and $l_e\mathbf{n}_e = -\mathbf{k} \times \mathbf{l}_e$ if $c$ is the first one in the list $c(e)$. Since the horizontal velocity appears as the product with the thickness, the expressions here can be

rewritten in terms of transports $\mathbf{U} = \mathbf{u}h$.

The fluxes through the top and bottom faces are computed with $w_c = \sum_{v(c)} w_v/3$ using either centered or the standard third-order upwind algorithms.

– **Scalar control volumes.** Instead of using vector control volumes, we assemble the flux divergence on the scalar control volumes and then average the result from the vertices to the cells. For the horizontal part,

$$A_v(\nabla \cdot (h\mathbf{u}\mathbf{u}))_v = \sum_{e(v)} \sum_{c(e)} \mathbf{u}_c h_c \cdot \mathbf{n}_{ec} \mathbf{u}_c d_{ec},$$

with the same rule for the normals as in the computations of the divergence operator. The contributions from the top and bottom faces of scalar control volume are obtained by summing the contributions from the cells:

$$A_v(w_v\mathbf{u}^t) = w_v \sum_{\overline{c}(v)} \mathbf{u}_c^t A_c/3$$

for the top surface, and similarly for the bottom one. The estimate of $\mathbf{u}^t$ can be either centered or third-order upwind as above. Other method will follow.

This option is special in the sense that the continuity is treated here in the same way as for the scalar quantities.

– **Vector-invariant form.**



The relative vorticity in the cell-vertex discretization is defined on vertices, and so should be the Coriolis parameter. We use the following representation

$$((\omega + f)\mathbf{k} \times \mathbf{u})_c = \sum_{v(c)} (\omega + f)_v \mathbf{k} \times \mathbf{u}_c / 3.$$

The representation with the thicknesses,

$$((\omega + f)\mathbf{k} \times \mathbf{u})_c = \sum_{v(c)} \frac{\omega_v + f_v}{3h_v} \mathbf{k} \times \mathbf{u}_c h_c$$

is reserved for future. The gradient of kinetic energy should be computed in the same way as the pressure gradient, which necessitates computations of $\mathbf{u}^2$ at vertices. This is done as

$$A_v \mathbf{u}_v^2 = \sum_{\overline{c}(v)} A_c \mathbf{u}_c^2 / 3.$$

The vertical part follows (**??**),

$$(w\partial_z \mathbf{u})_c^t = 2(\mathbf{u}_{(k-1)c} - \mathbf{u}_{kc})/(h_{(k-1)c} + h_{kc}) \sum_{v(c)} w_{kv} / 3$$

for the top surface and similarly for the bottom. Note that the contributions from the curl of horizontal velocity, the gradient of kinetic energy and the vertical part involve the same stencil of horizontal velocities.

The three options above behave similarly in simple tests on triangular meshes, but their effect on flow-topography interactions or eddy dynamics remains to be studied. The vector invariant option is slightly less dissipative, but may leave some noise in $w$ in areas where mesh resolution is varied (see Danilov and Wang (2015)) which is absent for the flux forms. Higher order methods can be applied for momentum flux computations, exploring which is reserved for future.

### 4.3 Viscosity operators

Formally, the derivatives of horizontal velocity can be estimated and the components of the viscous stress tensor, $\sigma_{ij} = \nu_h(\partial_i u_j + \partial_j u_i)$, can be found. Here the indices $i, j$ imply the horizontal directions, and $\nu_h$ is the horizontal viscosity. In computing their divergence a centered estimate of stresses has to be taken at the lateral faces of vector control volumes. If discretized in terms of cell velocities, this scheme downweights or fully eliminates the contributions from the nearest cells, and is thus incapable of eliminating grid-scale fluctuations in velocities.

The expression for stresses can be simplified as $\sigma_{ij} = \nu_h \partial_i u_j$. As discussed by Griffies (2004), its divergence still ensures energy dissipation, but is nonzero for solid-body rotations if $\nu_h$ is variable. However, for this expression, the contributions from the neighbor velocities in flux divergence can be strengthened by noting that only contraction with normal vector $\nu_h n_i \partial_i u_j$, i.e., the derivative in the direction of the normal $\mathbf{n}$, appears in the contributions for each vertical face. For the face





identified with edge $e$ between cells $c$ and $n$ we formally write $\mathbf{n} = \mathbf{r}_{cn}/|\mathbf{r}_{cn}| + (\mathbf{n} - \mathbf{r}_{cn}/|\mathbf{r}_{cn}|)$, where $\mathbf{r}_{cn} = \mathbf{d}_{en} - \mathbf{d}_{ec}$ is the vector connecting the centroids of cells $c$ and $n$, and split the stress

contracted with $\mathbf{n}$ into two respective parts. The velocity derivative (up to metric terms) in the direction of $\mathbf{r}_{cn}$ is just the difference between the neighboring velocities divided by the distance $|\mathbf{r}_{cn}|$. The remaining part of viscous flux (contracted with $(\mathbf{n} - \mathbf{r}_{cn}/|\mathbf{r}_{cn}|)$) is computed with the standard procedure based on centered estimate of stresses. It is easy to see that only the nearest neighbors will be involved on equilateral meshes (for $\mathbf{n}$ and $\mathbf{r}_{cn}$ are collinear). However, the computations of ve-

locity derivatives and stresses are still needed if meshes deviate from equilateral. The discretization of harmonic viscosity operator, amended as described above, works well. Its biharmonic version is obtained by applying the procedure twice.

This procedure, especially its biharmonic version, proves to be costly for it involves computations of velocity derivatives and manipulations with two types of contributions. On the other hand we see

that the expensive part involving the general computation of velocity derivatives is only needed on deformed meshes; it will be small on quasi-equilateral meshes and, even if it is not small generally, it contributes little to penalizing differences between the nearest velocities. This leads to the idea to introduce simplified operators based on the nearest neighbors. Indeed, by writing

$$A_c(D_u\mathbf{u})_c = (1/2)\sum_{e(c)}(\nu_{hn} + \nu_{hc})(\mathbf{u}_n - \mathbf{u}_c)l_e/|\mathbf{r}_{cn}|, \qquad (12)$$

where $n$ is the cell sharing edge $e$ with cell $c$, we take into account the contributions from the nearest neighbors. This expression is written for a uniform layer thickness, but can be adjusted for a variable one by adding $h_c$ on the lhs and $h_e$ on the rhs. The computation is implemented as a cycle over edges. One uses ghost velocities to impose boundary conditions, or can skip the contributions from the boundary edges to emulate free slip. It is easy to see that the operator integrates to zero in the

domain interior (momentum conservation) and is negative definite in the energy norm. The procedure is applied twice to get a biharmonic version.

The procedure can be simplified even further as

$$(D_u\mathbf{u})_c = \tau_u^{-1}\sum_{e(c)}(\mathbf{u}_n - \mathbf{u}_c). \qquad (13)$$

Here $\tau_u$ is a factor with dimension of time to be specified further. This variant is a filter removing

grid-scale fluctuations. Clearly, in a general case, it does not ensure momentum conservation, and we cannot strictly prove that it leads to kinetic energy dissipation. However, on equilateral triangular meshes it reduces to $D_u = (l^2/3\tau_u)(\partial_{xx} + \partial_{yy})$, where $l$ is the triangle side. This allows one to identify $\tau_u$ with $l^2/(3\nu_h)$. The biharmonic form of filter is taken as $-\tau_u D_u(A_0/A_c)^{1/2}D_u$, where $A_0$ is the reference cell area. In this case $\tau_u = l^3 l_0/(9\nu_{bh})$, where $l_0$ is the side of the reference cell

and $\nu_{bh}$ the coefficient of biharmonic viscosity. The inclusion of area scaling is needed for cubic dependence on $l$. Writing $\nu_{bh}$ in the commonly used form $\nu_{bh} = Vl^3$, where $V$ is the velocity scale,



one finds $V = l_0/9\tau_u$. The values about 0.02 m/s are generally sufficient even on highly variable meshes.

The code contains these options but we are using the last one in the biharmonic version in most cases — it is efficient both computationally and in terms of providing stable code performance. We have not met any visible artefacts thus far despite its obvious physical shortcomings. In all other cases, the coefficient of horizontal viscosity is scaled with mesh size to provide $\nu_h = Vl$ in the harmonic case and $\nu_h = (Vl^3)$ in the biharmonic case.

We note that the inefficiency of the standard Laplace operator in filtering grid scales for cell variable placement and measures needed to amend it are well known (see, e.g., Blazek (2001)). For the co-called ZM discretization, which is similar to the cell-vertex discretization up to the detail of scalar control volumes, Ringler and Randall (2002) proposed to introduce a small-stencil vector Laplacian operator based on the identity $\Delta\mathbf{u} = \nabla\nabla\cdot\mathbf{u} - \nabla\times\nabla\times\mathbf{u}$. The stencil involves only the nearest neighbors. However, because these computations are not related to the full mesh cells, they neither ensure momentum conservation nor negative definiteness of kinetic energy dissipation in a general case. In this respect using them is not more logical than using the simplified forms (12) or (13).

### 4.4 Transport of scalar quantities

High-order transport schemes for vertex variable placement can be realized by using polynomial reconstruction of scalar fields or the reconstruction of gradients of scalar fields at mid-edges. We experimented with the quadratic reconstruction of scalars, which provides a compromise between accuracy and computational effort (see Skamarock and Menchaca (2010)). Its other advantage for vertex placement of variables is that it needs only the information from the nearest neighbors, which imposes no new demands on halo exchange in parallel implementation. It turned out that it is not more accurate than the gradient reconstruction algorithm, being twice as expensive and demanding much more storage for the reconstruction matrices. For this reason, at present we keep the gradient reconstruction algorithm as the basic one, which is also available in combination with the FCT (flux corrected transport) algorithm.

Consider edge $e$ with $v(e) = (v_1, v_2)$ and $c(e) = (c_1, c_2)$. The advective flux of scalar quantity $T$ through the face of scalar volume associated to this edge is

$$F_e = T_e(-h_{c_1}\mathbf{d}_{ec_1}\times\mathbf{u}_{c_1} + h_{c_2}\mathbf{d}_{ec_2}\times\mathbf{u}_{c_2})\cdot\mathbf{k} = T_e Q_e.$$

The quantity $Q_e$ is the volume flux associated with edge $e$ which leaves the control volume $v_1$. We need an estimate for $T_e$ at the mid-edge. In order to provide it, for each edge $e$ we store the indices of the cells ahead or behind this edge in the direction $\mathbf{l}_e$. We compute two estimates

$$T_e^+ = T_{v_1} + (1/2)\mathbf{l}_e(\nabla T)_e^+, \quad (\nabla T)_e^+ = (2/3)(\nabla T)^c + (1/3)(\nabla T)^u,$$





and

$$T_e^- = T_{v_2} - (1/2)\mathbf{l}_e(\nabla T)_e^-, \quad (\nabla T)_e^- = (2/3)(\nabla T)^c + (1/3)(\nabla T)^d,$$

where the upper index $c$ implies the centered estimate, while $u$ and $d$ imply the gradients on up-edge and down-edge cells (computed by applying the stored scalar gradient operator). Since the centered estimate is only needed in the direction of edge, $\mathbf{l}_e(\nabla T)^c = T_{v_2} - T_{v_1}$. Taking

$$2T_e Q_e = (Q_e + |Q_e|)T_e^+ + (Q_e - |Q_e|)T_e^-$$

one obtains the standard third-order upwind method, and the estimate

$$2T_e = T_e^+ + T_e^-$$

provides the fourth-order centered method. The third-order method is a bit too dissipative, at least in eddy-dominated flows. The combination

$$2Q_e T_e = (Q_e + (1-\gamma)|Q_e|)T_e^+ + (Q_e - (1-\gamma)|Q_e|)T_e^-$$

takes the fourth-order part with the weight $\gamma$ and the third order part, with $1-\gamma$. In practice, $\gamma$ between 0.75 and 0.85 works well for many cases, reducing the upwind dissipation considerably (by a factor of 4 for $\gamma=0.75$). These are the recommended values.

We note that the high order of the scheme above is only achieved on uniform meshes. However, since $T_e$ is computed through linear reconstruction, the second order is warranted on general meshes.

The implementation requires preliminary computation of scalar gradients on cells. An extended halo exchange is needed to make these gradients available during flux assembly. Edges touching the topography may lack either $u$ or $d$ cells. In this case the simplest choice is either to use the central estimate or the estimate based on the mean vertex gradient $A_v(\nabla T)_v = \sum_{\bar{c}(v)} A_c(\nabla T)_c/3$. This introduces some additional logistics, but it is common for all high-order schemes.

For the vertical direction, we provide a set of possibilities which include the third/fourth order option similar to the described above, spline interpolation as well as the piece-wise parabolic method by Colella and Woodward (1984).

The FCT version uses the first-order upwind method as the low-order monotonic method and the method above as the high-order one. The low-order solution and the antidiffusive fluxes (the difference between the high-order and low-order fluxes) are assembled in the same cycle (over edges for the horizontal part and over vertices for the vertical part) and stored. We experimented with separate pre-limiting of horizontal and vertical antidiffusive fluxes and found that commonly this leads to an increased dissipation, for the horizontal admissible bounds are in many cases too tight. For this reason, the computation of admissible bounds and limiting is three-dimensional. As the result, it will not necessarily fully eliminate non-monotonic behavior in the horizontal direction. The FCT algorithm of FESOM1.4 follows the same logics, however, in that case it is the only possibility. Using the FCT version roughly doubles the cost of transport algorithm, but adds stability needed in practice.





### 4.5 Vertical velocity splitting

As demonstrated in Lemarié et al. (2015), in practice, the strongest Courant number limitation comes from vertical advection in isolated patches adjacent to the coast. The code numerical efficiency can be augmented if some measures are taken to stabilize it with respect to vertical advection. Unstructured meshes of variable resolution might be even more vulnerable to such limitation because their irregularity can easily provoke a noisy pattern in $w$ through rendering of topography. We implement the approach proposed by Shchepetkin (2015) according to which the vertical transport velocity is split into two contributions $w = w_{ex} + w_{im}$ where the first one is determined by the maximum admissible Courant number, and the second one is the rest. The advection with $w_{ex}$ is done explicitly using schemes mentioned above. The advection with $w_{im}$ is implicit. It uses the first-order upwind (backward Euler in time) so that the vertical operator that corresponds to it is diagonally dominant. The latter is solved together with the implicit vertical mixing by the standard sweep algorithm. As a result, if this option is used, the incurring additional costs of the model time step are negligible. The use of the first order upwind scheme may seem to be too dissipative, but the point is that it is applied only to the part of velocity and only in critical cases.

### 4.6 GM and isoneutral operators

#### 4.6.1 The eddy-induced transport

There are several ways to implement the Gent–McWilliams (GM) parameterization (Gent and McWilliams (1990); Gent et al. (1995)). We follow the algorithm proposed by Ferrari et al. (2010) in FESOM2. FESOM1.4 operates with skewsion (see Griffies (2004) for mathematical detail).

The bolus velocity $\mathbf{v}^* = (\mathbf{u}^*, w^*)$ is expressed in terms of eddy-induced streamfunction $\mathbf{\Psi}$,

$$\mathbf{v}^* = \nabla_3 \times \mathbf{\Psi}, \quad \mathbf{\Psi} = \boldsymbol{\gamma} \times \mathbf{k},$$

where $\boldsymbol{\gamma}$ is a two-dimensional vector. Ferrari et al. (2010) suggest to compute it by solving

$$(c^2 \partial_{zz} - N^2)\boldsymbol{\gamma} = (g/\rho_0)\kappa \nabla_z \sigma \tag{14}$$

with boundary conditions $\boldsymbol{\gamma} = 0$ at the surface and ocean bottom. In this expression, $c$ is the speed of the first baroclinic mode, $\sigma$ the isoneutral density, $\kappa$ the thickness diffusivity, $N$ the Brunt–Väisälä frequency, and the index $z$ means that the gradient is computed for fixed $z$ (it differs from the gradient along layers, $\nabla_z \sigma = \nabla \sigma - \partial_z \sigma \nabla Z$). In terms of the vector $\boldsymbol{\gamma}$ the components of eddy-induced velocity are computed as

$$\mathbf{u}^* = \partial_z \boldsymbol{\gamma}, \quad w^* = -\nabla \cdot \boldsymbol{\gamma}.$$

It is easy to see that solving (14) plays a role of tapering, for it allows one to smoothly satisfy boundary conditions. Because of the boundary conditions adding the eddy-induced velocity to the mean velocity $(\mathbf{u}, w)$ does not change $\overline{h}$ as the vertically integrated divergence of $\mathbf{u}^*$ is zero. In





the ALE formulation the inclusion of eddy-induced velocity implies that the thickness and tracer equations are now written for the so called residual velocity $\mathbf{u}_r = \mathbf{u} + \mathbf{u}^*$, $w_r = w + w^*$.

Although the natural placement for $\boldsymbol{\gamma}$ is at the cell centroids, we solve for it on the mesh vertices in order to reduce the amount of computations. The vertical location is at full levels (layer interfaces). The horizontal bolus velocities are then computed at cell centroids as

$$\mathbf{u}_c^* = (1/3)\partial_z \sum_{v(c)} \boldsymbol{\gamma}_v.$$

The vertical bolus velocity $w^*$ is then found together with $w$ at the end of the ALE step and the full residual velocity is used to advect tracers.

We compute the speed $c$ in the WKB approximation as

$$c = \frac{1}{\pi} \int\limits_{-H}^{0} N dz.$$

Among other factors, the magnitude of the thickness diffusivity $\kappa$ depends on the resolution $r$ and the local Rossby radius $L_R = c/f$:

$$\kappa = \kappa_0 f_\kappa(r/L_R),$$

where $f_\kappa$ is a cut-off function that tends to 0 if $r/L_R < 1$ and to 1 otherwise. The resolution is defined as a square root of the area of the scalar control volume. On general meshes it may exhibit substantial local variations, so smoothing over the neighboring vertices is done. Note that scaling

with mesh resolution for viscosity and diffusivity coefficients will also benefit from using a smoothed $r$.

#### 4.6.2 Isoneutral diffusion

Assuming that the slope of isopycnals is small, we can write the diffusivity tensor as

$$\mathbf{K} = \begin{pmatrix} K_i & 0 & s_x K_i \\ 0 & K_i & s_y K_i \\ s_x K_i & s_y K_i & s^2 K_i + K_d \end{pmatrix}. \tag{15}$$

Here $K_i$ and $K_d$ are the isoneutral and diapycnal diffusivities, and $\mathbf{s}$ is the isoneutral slope vector computed along layers,

$$\mathbf{s} = (s_x, s_y) = -\nabla\sigma/\partial_z\sigma.$$

If layers interfaces deviate substantially from geopotential surfaces, for example, if layers follow the bottom topography, the slope vector can be substantially larger than typically found on $z$-coordinate meshes. Mixed derivatives in $\nabla_3 h \mathbf{K} \nabla_3$ operator in this case can cause time step limitations (Lemarié et al. (2012a)). To maintain stability, the term $h\partial_z(s^2 K_i + K_d)\partial_z$ has to be treated implicitly. Appendix D shows the details of the numerical implementation of isoneutral diffusion.



## 5 FESOM2.0 versus FESOM1.4

In the following we evaluate the performance of FESOM2.0 by simulating the realistic ocean state under prescribed atmospheric forcing. The purpose is to illustrate that FESOM2.0 is ready to be run in global configurations, although it still may need some further parameter tuning. Model efficiency is then briefly assessed. Detailed model assessment is the subject of future work.

### 5.1 Meshes

The evaluation will be done in two steps. In the first step we compare the performance of FESOM2.0 to that of finite-element FESOM1.4 (Wang et al., 2014). For this purpose, we run both models on the same coarse-resolution reference mesh and in similar configurations. The z-coordinate in the vertical is used in simulations described below. Although the same mesh and level surfaces are used, vertical mesh geometry is different: FESOM2.0 assumes the mesh to be composed of prisms whereas these prisms are split into tetrahedra in FESOM1.4. The mesh contains about 120,000 surface nodes, its horizontal resolution varies from 25 km in high latitudes of the Northern Hemisphere to nominally one degree elsewhere, and there are 46 unevenly spaced z-levels in the vertical. This mesh was also used to carry out FESOM1.4 simulations for model intercomparison in the Coordinated Ocean-ice Reference Experiments - Phase II (CORE-II Large and Yeager, 2009) project. It has been demonstrated that FESOM1.4 performs well in this configuration compared to other ocean models (see, e.g., Griffies et al. (2014); Danabasoglu et al. (2014) and other papers in the same virtual issue).

In the second step we simulate the ocean state under CORE-II forcing with FESOM2.0 but on an eddy-permitting global mesh with a quasi-uniform resolution of 15 km, referred to further as Glob15. The mesh contains about 2,000,000 surface nodes. It is worth mentioning that the size of Glob15 is already larger than all meshes we used with FESOM1.4 thus far. We did not carry simulations on Glob15 with FESOM1.4 to save computational resources.

### 5.2 Model settings

Although we try to configure both model versions as close as possible for our intercomparison, there are a few differences due to the details of implementation. First, different transport schemes are used. The Taylor-Galerkin (TG) algorithm of FESOM1.4 with consistent mass matrices is expected to be less dissipative than the third-fourth order upwind algorithm used in FESOM2.0. The TG scheme works by default with a FCT limiter in FESOM1.4, so we apply the FCT limiting in FESOM2 too. Second, the difference between the two versions of FESOM comes from the implementation of the GM parameterization of eddy transport. FESOM1.4 uses the GM skew flux formulation as suggested by Griffies (1998). Because of the finite-element discretization and hence variational formulation, this strategy is optimal for FESOM1.4 but less convenient for FESOM2.





All simulations are run with the linear free-surface and virtual salinity forcing. The surface salinity is restored to the climatological data with the piston velocity of 50 m/300 days which is a common
practice for stand-alone ocean models. Although the default mixing scheme in FESOM1.4 is the k-profile parametrization (KPP, Large et al., 1994), it has not been tested yet with FESOM2.0. That is why the vertical mixing in all simulations presented further is provided by the Pacanowski and Philander (1981) scheme with the background vertical diffusion of $2 \cdot 10^{-3}$ m$^2$ s$^{-1}$ for momentum and $10^{-5}$ m$^2$ s$^{-1}$ for the potential temperature and salinity, and the maximum is limited to 0.01 m$^2$
s$^{-1}$. The parameterization of mesoscale eddies was switched off in the simulation with Glob15 as suggested by Delworth and Coauthors (2012). The time step was set to 30 min and 15 min for the reference and Glob15 meshes, respectively in order to meet the CFL condition. All runs are initialized in winter from the Polar Science Center Hydrographic Climatology (Steele et al., 2001) and the integration covers the time frame 1948-2007 of CORE-II atmospheric forcing (Large and
Yeager, 2009).

### 5.3 Intercomparison on the coarse-resolution reference mesh

We first compare the last 15 years of the simulated hydrography in the two model runs on the coarse-resolution reference mesh to the World Ocean Database 2005 (WOA2005, Conkright et al., 2002). One should keep in mind that the spin-up time of 60 years is much too short to provide an equili-
brated ocean state. Nevertheless, the departure of the modeled hydrography from climatology after 60 years of integration can already serve as a measure of the model drift and indicate the quality of solution. The bias of temperature in different depth ranges is shown in figure 3 for FESOM1.4 and FESOM2.0, respectively. The patterns for the upper 200m look generally similar in the models. Notably, for the cold bias in the Labrador Sea, its surrounding and in the region of Malvinas Current,
FESOM2.0 simulates much larger departures from WOA2005 than FESOM1.4. This cold bias is primarily associated with the missing northwest corner in the path of the North Atlantic Current and too weak strength of the subpolar gyre. Both are often attributed to the lack of spatial resolution. Marzocchi et al. (2015) show that the resolution of 1/12° (on ORCA type meshes) is already sufficient to properly resolve the pathway of the North Atlantic Current. On other hand, at coarse resolutions,
Stouffer et al. (2005) and Jochum et al. (2008) demonstrate that the reduction in viscosity in the extratropical ocean in climate models increases the strength of subpolar gyre in the North Atlantic. Other experiments carried out with FESOM, which are not presented here, indicate that even small changes in model parameters like viscosity and GM thickness diffusivity can impact the strength of the cold bias. Improving the simulation quality in the Labrador Sea and its vicinity by both, increas-
ing the local resolution and tuning the model parameters, will be the focus of future studies. It should be mentioned that the bias in the upper ocean hydrography shown here for FESOM1.4 is different from that presented in the intercomparison of CORE-II hindcasts (Griffies et al., 2014; Danabasoglu



et al., 2014) where the KPP mixing scheme was used instead of PP. As mentioned, different mixing schemes besides PP still need to be more thoroughly tested with FESOM2.0.

At deeper levels of the tropical Atlantic, FESOM2.0 performs better than FESOM1.4; at the same time, error become larger in the Southern Ocean and easter North Atlantic. Our experience in running FESOM is that the drift in the Southern Ocean is substantially affected by the imposed spatial (horizontal and vertical) pattern of the GM coefficient $\kappa$, which needs to be tuned in FESOM2. The warm bias in the eastern North Atlantic is a persistent feature in all simulations with FESOM2.0

and is likely due to an overly strong Mediterranean Outflow. A closer look at salinity (Fig. 4) reveals that FESOM2.0 simulates much fresher Mediterranean Sea than FESOM1.4 and more salt is released into the North Atlantic across the Strait of Gibraltar. Indeed, the meshes used here have an artificially widened strait, whereas the cell placement of velocity vectors and the free slip boundary condition applied in FESOM2.0 (no slip option used in FESOM1.4) have a potential to increase the

Gibraltar outflow if the same geometrical boundary is used. Although the idea of resolving the Strait of Gibraltar may seem straightforward, too fine resolution would lead to additional computational burden associated with a sufficiently small time steps. Individual adjustment of mesh geometry is required for two model versions.

The streamfunction of meridional overturning circulation (MOC) shown in figures 6a and 6b for

reference runs with FESOM1.4 and FESOM2.0, respectively, reveals that the Antarctic bottom water (AABW) production is larger in FESOM2.0 compared to FESOM1.4 and is at the upper boundary of the observation-based estimate available from the literature (see eg. Lumpkin and Speer, 2007). The maximum overturning of Upper Circumpolar Deep Water (UCDW) at about 35°S exceeds 20Sv compared to only 5Sv in FESOM1.4. This behavior suppresses the mid-depth cell at about 30°S. The

maximum of the mid-depth cell in the North Atlantic is about 12Sv in both versions and remains at the lower boundary of observational estimates published in the literature. Another distinction between both MOCs is at the northern boundary of the domain. As mentioned in Sidorenko et al. (2009) there is an ambiguity in transport definitions for discretizations exploiting the finite-element approach. This results in a bias that accumulates in the diagnosed MOC at the northern boundary

when integrating from the south to the north. The inconsistency amounts to about 2 Sv at certain depths in FESOM1.4 while it is zero in FESOM2.0. We conclude that FESOM1.4 and FESOM2.0 show similar behavior on the reference mesh, but FESOM2.0 may benefit from further tuning. In particular, the impact of vertical transport schemes, bottom representation and boundary conditions needs to be explored in more detail.



### 5.4 Eddy-permitting global simulation at 15 km resolution

#### 5.4.1 Simulated ocean state

The difference of hydrography simulated on Glob15 compared to WOA2005 is shown for the mean over the last 15 years in Fig. 3 and Fig. 4 (the right columns) for temperature and salinity, respectively. Overall, the model drift in Glob15 is smaller than in the reference runs. The largest improvement is seen at the surface, where the cold bias north of 45°N is now confined to the northwest corner of the North Atlantic Current. It does not vanish completely, however, because the resolution of 15 km is far from being even eddy permitting in this region, where the Rossby radius of deformation goes well below 10 km. The area with freshwater bias north of the Newfoundland has been significantly reduced compared to the reference simulation with FESOM2.0. This points to the improved linkage between Arctic and the North Atlantic oceans. Some other improvements are also seen at other locations and in different depth ranges as well. For instance, the bias in the Southern Ocean is remarkably reduced in the deeper layers as is visible from salinity patterns (Fig. 4). These improvements indicate that over some parts of the global ocean partially resolving mesoscale features can already impact dynamics.

In order to illustrate the eddy activity we show the snapshot of surface relative vorticity in the North Atlantic in Fig. 5. Although we show the North Atlantic only, the dynamics in Glob15 is eddy rich around all key fronts and in subtropial gyres of the global ocean. As expected, the mesoscale features are prominent in Fig. 5 along the Florida Current, Gulf Stream and the North Atlantic Current. The Azores Current branching off the Gulf Stream at ca. 35°N is reproduced also well. At higher latitudes above about 50°N the resolution becomes insufficient for capturing eddy dynamics because the Rossby radius decreases. In the high-resolution experiment with FESOM2.0 the Gulf Stream separates too far north of Cape Hatteras, a feature shared by most ocean models with the resolution below 0.1°. As one would expect, the wrong separation of Gulf Stream, is also reflected in the drift of hydrography, where too warm and salty bias develops close to the western coast.

The pattern of relative vorticity also reveals the existence of zonally elongated patches corresponding to zonal jets which are often simulated with the high resolution ocean models, and are confirmed by the altimetric observations (see eg. Maximenko et al., 2005). The stripes in the vorticity are seen primarily in the North and South Pacific and in the South Atlantic oceans (not shown). In the North Atlantic zonal jets are most visible at about 30°N and in the eastern NA at about 50°N. Note that for better visualization of zonal jets one shall inspect the vorticity pattern averaged over certain periods of time or do the same for zonal velocity component.

The MOC for Glob15 is shown in Fig. 6c and depicts significant improvements compared to that simulated with FESOM2.0 on the reference mesh. While the bottom cell has been reduced there is a significant increase in the mid-depth cell reaching a maximum of above 15Sv in the NA. The Antarctic Bottom Water export across about 65°S shows a clear connection with the UCDW



matching the estimates from inverse techniques by Lumpkin and Speer (2007). The reader is referred to Fig. 2 in their paper. The improvements seen for simulations on mesh Glob15 compared to the reference mesh may serve as an argument in favor of using high resolution.

### 5.4.2 Sea Ice

The sea ice thickness simulated on Glob15 is shown in Fig. 7 for March and September. The maps of ice thicknesses compare well to those of the Pan Arctic Ice-Ocean Modeling and Assimilation System (PIOMAS; Schweiger et al., 2011) presented in (Notz et al., 2013, their Fig. 8) for the Northern Hemisphere. The thickest sea ice in the Arctic reaches above 5m in March and September and is found north of Greenland and in the Canadian Archipelago, becoming thinner towards the Siberian

coast. The simulated 15% sea ice concentration contours, indicating the sea ice edge, are also shown in Fig. 7 (white contour line) together with NSIDC observations (Fetterer et al., 2002, updated 2009) (black contour line). In September, the model overestimates the sea ice coverage along the Siberian Shelf and in the northern Barents Sea. Because of this, the summer Arctic sea ice extent in Glob15 is on the average overestimated by 10% compared to the satellite data, providing $7.54 \cdot 10^6 km^2$

compared to $6.74 \cdot 10^6 km^2$ from NSIDC. In the Southern Hemisphere, Glob15 underestimates the summer ice extent. In this context further study of the performance of mixed-layer parameterization and the effect of still insufficiently strong eddies on the properties of the watermasses simulated around the Antarctic coast may be needed. The sea ice extent simulated by the new model version is very similar to that simulated by FESOM1.4, which lies within the spread of the CORE-II multi-

model ensemble (Downes et al. (2015)., Wang et al. (2016b)). This similarity is probably not too surprising given that both versions of FESOM share the same sea-ice component.

In order to quantify the seasonal variability of the sea ice we plot the monthly time series of sea ice extents in Fig. 8. The result compares well to the observation in the Northern Hemisphere, while the amplitude of seasonal variability is overestimated in the Southern Hemisphere. The model simulates

lower summer and higher winter sea ice extent in the Southern Ocean. For both hemispheres, the model captures realistic trends in sea ice extent from 1979 to 2007, which are negative and positive in the Northern and Southern Hemispheres, respectively.

### 5.5 Performance and implementation issues

FESOM is written in Fortran 90 with some C/C++ code inserts providing bindings to the third party

libraries. The code employs the distributed memory parallelization based on MPI (Message Parsing Interface). The model experiments have been carried out on a Cray XC40 system hardwared with Intel Xeon Haswell and 24 cores per node, which was made available through the North-German Supercomputing Alliance (HLRN). The experience shows that the parallel scalability of both versions of FESOM starts to saturate after assigning less than 300 vertices of surface mesh per computational

core. In view of this, the experiments on the reference were conducted using 384 cores (16 nodes).



Disregarding input/output, the throughput of FESOM1.4 is ca. 25 simulated years per day (SYPD) where 92.5% and 7.5% of the resources are spent in the ocean and ice components, respectively. The resources spent in dynamical (solving for $u, v, w, \eta$) and tracer (solving for $T, S$) parts in the ocean are nearly equal. The performance of the dynamical part of the ocean component highly relies on

the numerical solver used to solve for the external mode (elevation). We use the parallel Algebraic Recursive Multilevel Solver (pARMS, Li et al., 2003) augmented with Schur Complement Preconditioner with local Incomplete LU-Factorization (Fuchs, 2013). The cost of solving with pARMS is only about 10% of the dynamical part and nearly 5% of the total cost.

Using the same computer resources, the throughput of FESOM2.0 is 110 SYPD. In this version,

the resources between the ocean and sea ice components are split as 67% and 33%, respectively. The ocean component in FESOM2.0 demonstrates a 7 times higher throughput than that of FESOM1.4 giving the largest speedup in the tracer part, where it is even 9 times faster than in FESOM1.4. The implementation of GM after Ferrari et al. (2010) costs nearly 10% in the ocean component and 20% is spent in pARMS to solve for the sea surface height. Interestingly, pARMS shows much faster

convergence (up to a factor of 2.5) in FESOM2.0 than in FESOM1.4. In summary, disregarding input/output, the reference setup FESOM2.0 shows about 5 times higher throughput than FESOM1.4.

The Glob15 configuration was run on 1728 cores (72 nodes) giving a throughput of 17 SYPD, with relative costs between model components remaining comparable to those of the coarser-resolution reference setup. For this mesh the relative cost of using pARMS decreases compared to the reference

mesh despite the much larger mesh and the number of cores. We guess that it is partly linked to a smaller time step which improves the diagonal dominance in the matrix of sea surface height operator. Compared to the reference mesh, which was run in the limit of linear scalability ($\approx$300 surface vertices per core), Glob15 was run with $\approx$1150 vertices per core, so there is still potential for further increase in troughput.

The numbers given above serve only to illustrate the computational performance. Details may depend on the frequency of output, the type of transport algorithm, the presence of isoneutral diffusion or GM parameterization and the number of subcycles used in the Elastic-visco-plastic sea-ice solver of FESIM (Danilov et al., 2015)). A conservative estimate would be a three-fold speedup compared to FESOM1.4.

## 6 Discussion

### 6.1 From finite elements to finite volumes

There are several reasons for developing a new dynamical core based on finite-volume discretization. The first and the main one is the need for enhanced numerical efficiency. Generally, the codes based on unstructured meshes are less efficient numerically than their structured-mesh counterparts

partly because of (i) indirect indexing and the need for numerous auxiliary (look-up) arrays (neigh-




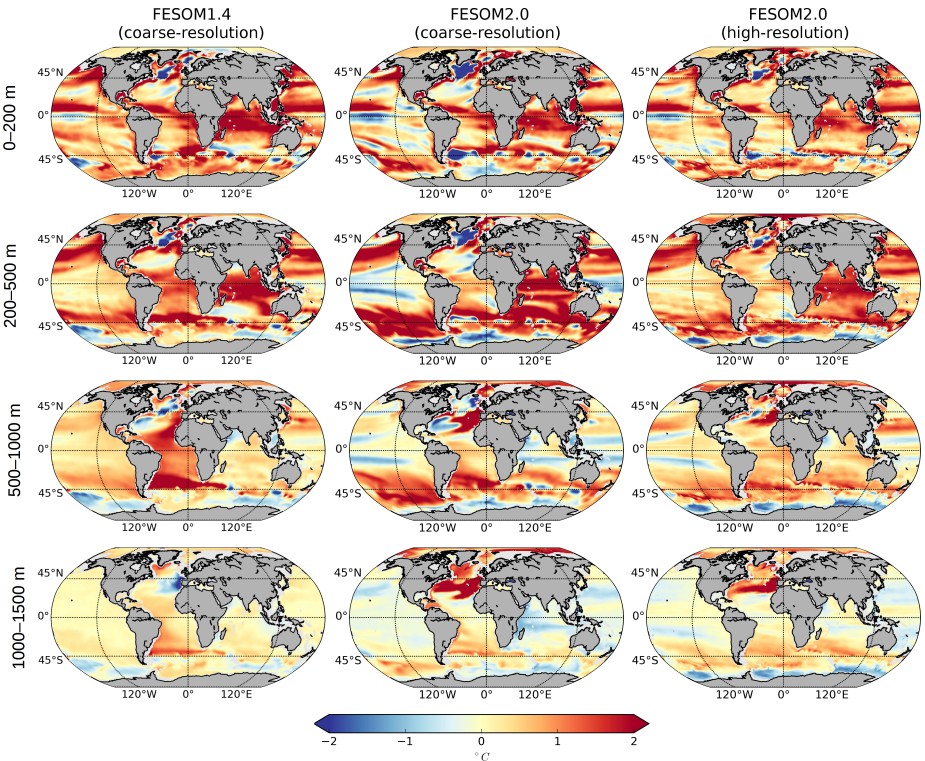

Figure 3: The departure of simulated potential temperature averaged over 1998-2007 from WOA2005 climatology, averaged over depth ranges. The left and middle columns correspond to the simulations performed with FESOM1.4 and FESOM2 respectively on the coarse-resolution reference mesh. The right column corresponds to FESOM2.0 on the global mesh with resolution of 15 km (no GM parameterization is used in this case).

boring cells, vertices, matrices of horizontal derivatives) and partly because of (ii) increased share of floating-point and memory-access operations needed in the absence of directional splitting and mesh structure. The overhead related to (i) can be minimized in codes using prismatic elements defined by unstructured surface meshes. In this case the same 2D auxiliary arrays can be used over the entire

water column, which makes the cost of assessing them rather moderate. The overhead of (3D) auxiliary arrays is much larger in FESOM1.4 because of its tetrahedral elements needed to implement arbitrary level surfaces. Using bilinear prismatic elements (Wang et al. (2008)) requires to store and access Jacobians on generalized meshes, which adds to the computational burden. Turning to the finite volume method together with the ALE vertical coordinate provides a simple and efficient way

to exploit the benefits of prismatic meshes.




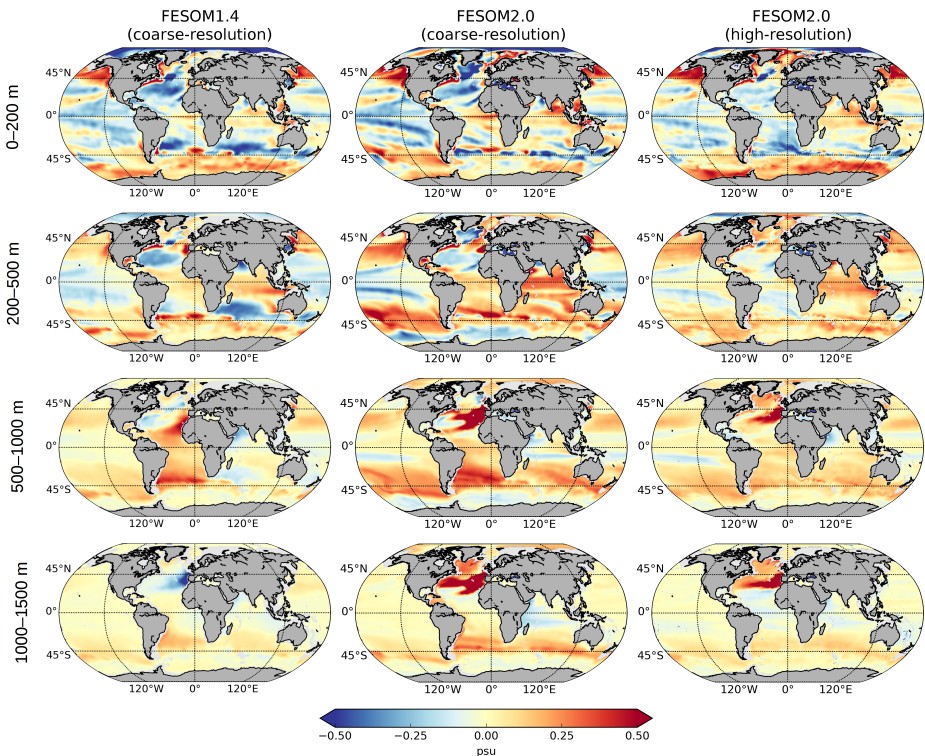

Figure 4: The same as in fig. 3 but for salinity.

The second reason for switching to a finite volume dicretization is that, as mentioned in Danilov (2013), continuous Galerkin finite elements are suboptimal for hydrostatic codes because of creating horizontal connections even in the matrices of purely vertical operators. In order to be practical, FE-SOM1.4 used a potential $\phi$ for the vertical velocity $w = \partial_z \phi$ and finite difference method to compute

pressure from the hydrostatic balance. This destroys energetic consistency between conversions of kinetic and available potential energy. The finite-volume discretization allows us to maintain energetic consistency (up to errors due to temporal discretization).

Finally, the finite-volume discretization operates with clear definition of fluxes, which is much more convenient for post-processing. For example, it makes computations of the meridional over-

turning streamfunction much more straightforward and free of interpretation inconsistencies intrinsic to the continuous finite-element discretization. In addition, it also allows numerous transport algorithms, whereas the choice available for finite-elements of selected type is much more restrictive.



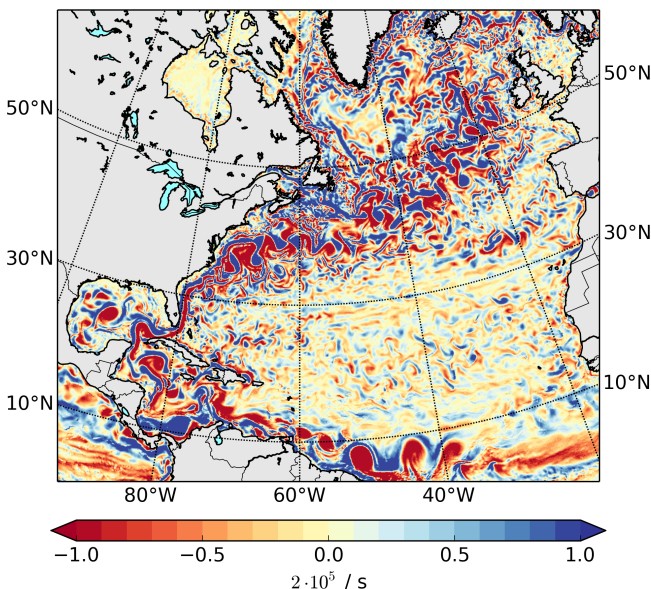

Figure 5: A snapshot of subsurface (40m) relative vorticity for the 1 January 2007 from eddy-permitting simulation with FESOM2.0 on the global 15 km mesh.

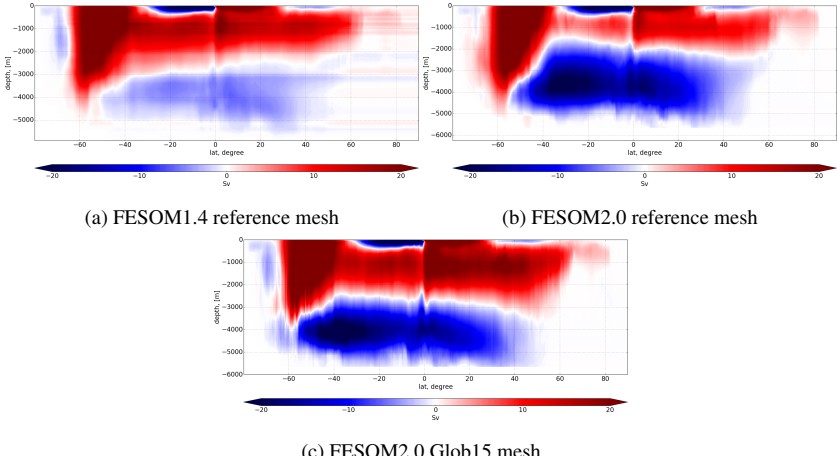

(a) FESOM1.4 reference mesh        (b) FESOM2.0 reference mesh

(c) FESOM2.0 Glob15 mesh

Figure 6: Eulerian-mean meridional overturning streamfunction averaged over the last 15 years of 60-year simulations for FESOM1.4 on the reference mesh (a) and FESOM2.0 on the reference (b) and Glob15 (c) meshes.





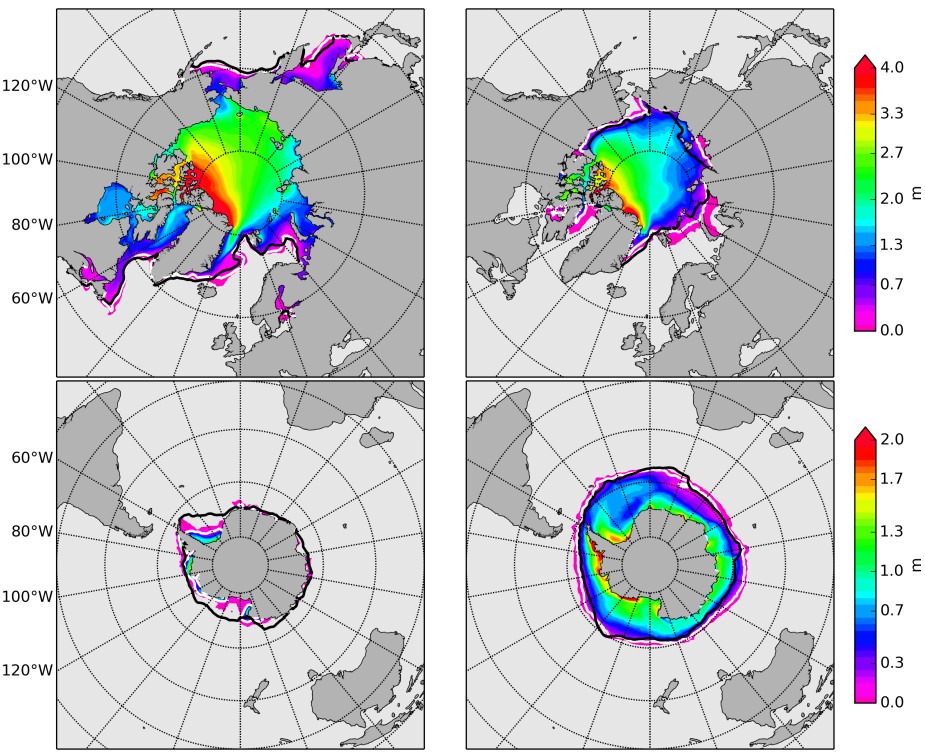

Figure 7: The simulated mean ice thickness distribution (m) in the northern (top) and southern (bottom) hemispheres in March (left) and September (right).

## 6.2 Cell-vertex discretization

Among possible finite volume discretizations the cell-vertex discretization used by FESOM2 presents
a compromise allowing us to keep general triangular meshes and use staggering of velocities and
pressure. A collocated vertex-vertex finite-volume discretization, which is the closest analog to FE-
SOM1.4, was explored by Danilov (2012). It presents a finite-volume analog of linear finite elements,
and needs stabilization on uneven bottom against pressure modes for the same reason as FESOM1.4.
Although stabilization does not necessarily lead to deficiencies in the simulated ocean state, it intro-
duces biases to energy exchanges and geostrophic balance, which should better be avoided. Addi-
tionally, it requires to split the horizontal velocities into contributions located on vertices and cells,
so that the velocity used to transport scalar quantities and the velocity used to compute momen-
tum balance are different entities. The cell-vertex discretization is free of pressure modes; however,
this comes at the price of an excessively large number of velocity degrees of freedom. This creates
spurious inertial velocity modes and requires the presence of efficient grid-scale viscosity operator





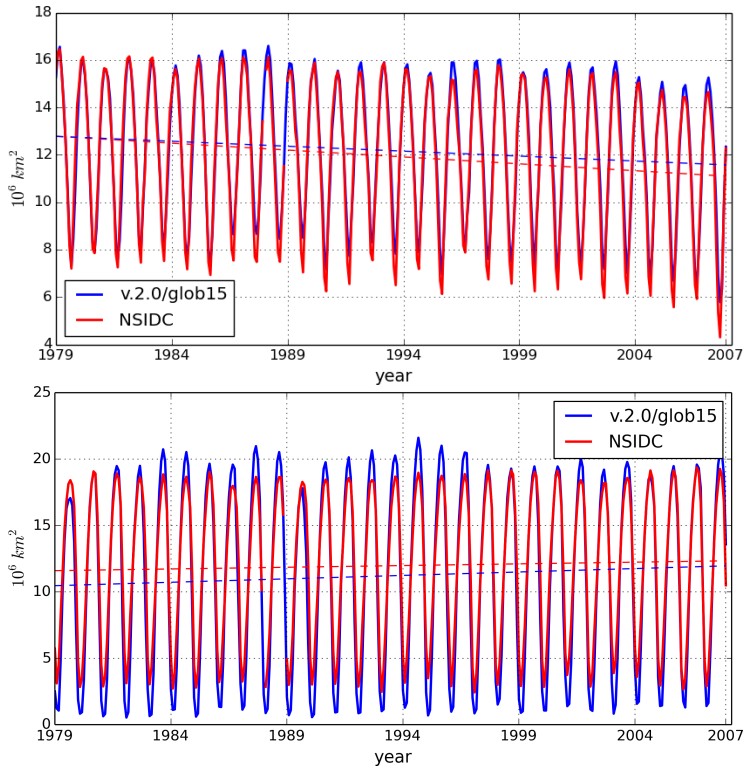

Figure 8: The simulated ice extent in the northern (top) and southern (bottom) hemispheres

coupling neighboring velocities. We have found that biharmonic filters are efficient in accomplishing this even on highly nonuniform meshes.

Because of staggering and keeping the velocity vector, the triangular cell-vertex discretization is an analog of an inverted B-grid (we call it quasi-B-grid). The inversion (the domain boundary is defined by scalar points) allows us to implement both free- and no-slip boundary conditions. Spurious inertial modes are absent on quadrilateral B-grids. This prompts us to consider hybrid meshes composed of triangles and quads, where the triangles will be used to provide transitions between regions of different resolution. The generalization to hybrid meshes is straightforward in the finite-volume implementation because most of operations are implemented as a cycle over edges. Furthermore, since the number of edges on quadrilateral meshes is smaller than on triangular meshes for a given number of vertices, this also implies a speed up in the code performance. This strategy is already implemented in the coastal branch of FESOM (to be described elsewhere) and will be made available in FESOM later.



Two other variants of finite-volume discretization are used at present in global ocean circulation
models. MPAS (Ringler et al. (2013)) uses a C-grid discretization on the Voronoi polygonal meshes
(most of polygons are hexagons), and the ICON implementation (at the Max Planck Institute for Me-
teorology, Hamburg) is based on a triangular C-grid (which needs an orthogonal triangular mesh).
The spurious modes of hexagonal C-grid are well controlled, but hexagons are less flexible geomet-
rically and were not selected for our development. The triangular C-grids have spurious divergence
modes which seem to be more difficult to control than inertial modes of cell-vertex discretization.
Practical experience gained in future through using models with different types of unstructured-mesh
finite-volume discretization will reveal the most efficient choice. The community effort may lead to
certain convergence among future model versions, similarly to the convergence toward C-grids ob-
served presently for models formulated on structured quadrilateral meshes.

## 7 Conclusions

This paper describes version 2 of FESOM. The new numerical core uses a cell-vertex finite-volume
discretization. FESOM2.0 compares well with FESOM1.4 in terms of simulated global ocean circu-
lation. It inherits the model framework and the sea ice model of its predecessor, and is conceived so
as to allow users familiar with FESOM1.4 to switch the versions easily. FESOM2.0 ensures higher
numerical throughput than FESOM1.4, which makes it much closer to the structured-mesh models
in terms of numerical efficiency. It offers new functionality through the ALE vertical coordinate. Fu-
ture development will focus on the generalized vertical coordinates, high-order transport algorithms
working on partly terrain-following meshes without excessive diapycnal mixing and on generaliza-
tion to mixed meshes combining triangles and quads. FESOM2 will gradually replace FESOM1.4,
yet the latter will be maintained and users support will be provided over several years to come.

## 8 Code and data availability

The version of FESOM2.0 used to carry out simulations reported here can be accessed from
https://swrepo1.awi.de/svn/awi-fvom/ after registration. The updated versions will be available through
the same link in future. For convenience, the configuration used, together with the meshes, is archived
at https://doi.org/10.5281/zenodo.161319. Mesh partitioning in FESOM is based on a METIS Ver-
sion 5.1.0 Package developed at the Department of Computer Science & Engineering at the Univer-
sity of Minnesota (http://www.cs.umn.edu/~metis). METIS and pARMS (Li et al., 2003) present
separate libraries which are freely available subject to their licenses. FESOM1.4 is available at
https://swrepo1.awi.de/projects/fesom/ (requires registration). The Polar Science Center Hydrographic
Climatology (Steele et al., 2001) used to initialize runs CORE-II atmospheric forcing data (Large and
Yeager, 2009) are freely available online. The simulation results can be obtained from the authors
on request.





**Acknowlegements** We are indebted to our colleagues N. Rakowski and S. Harig for their support
and help with numerous details. We also acknowledge the contribution of K. Korchuk at early stages
of FESOM2.

### Appendix A:  The flux form of momentum advection

When using the flux form of momentum, the natural choice is $h^* = h^n$, which makes the thickness
and transport equations centered. The choice for the thickness appearing with pressure is $h^{n+1/2}$,
which is centered. The advection and Coriolis terms will be computed through AB2 (or AB3) time
stepping, or if needed, the Coriolis term can be made semiimplicit. The transport $\mathbf{U}^n = \mathbf{u}^n h^n$ be-
comes a natural velocity variable.

The time stepping algorithm can be formulated as follows

$$\mathbf{U}^{n+1} - \mathbf{U}^n = \tau(\mathbf{R}_U^{n+1/2} - gh^{n+1/2}\nabla(\theta\eta^{n+1} + (1-\theta)\eta^n) + (\nu_v\partial_z\mathbf{u}^{n+1})^t - (\nu_v\partial_z\mathbf{u}^{n+1})^b)$$

with

$$\mathbf{R}_U^{n+1/2} = (\mathbf{R}_U^*)^{AB} - h^{n+1/2}(\nabla p_h + g\rho\nabla Z)/\rho_0,$$

and

$$\mathbf{R}_U^* = -\nabla\cdot(\mathbf{U}^n\mathbf{u}^n) - (w^t\mathbf{u}^t - w^b\mathbf{u}^b)^n - f\mathbf{k}\times\mathbf{U}^n.$$

The last expression combines the terms that need the AB method for stability and the second order.
We use $h^{n+1/2}$ to compute $Z$ and follow the same rule as (7) to compute $\eta^n$. The steps are:

– Do the predictor step and compute $\Delta\tilde{\mathbf{U}} = \tau\mathbf{R}_U^{n+1/2} - \tau gh^{n+1/2}\nabla\eta^n$.

– Update for implicit viscosity.

$$\partial_t\Delta\mathbf{U} - (\nu_v\partial_z(\Delta\mathbf{U}/h^{n+1/2}))|_b^t = \Delta\tilde{\mathbf{U}} + (\nu_v\partial_z(\mathbf{U}^n/h^{n+1/2}))|_b^t.$$

– Solve for new elevation. We write first

$$\overline{\mathbf{U}} = \sum_k \mathbf{U},$$

and similarly for other quantities, getting

$$\overline{\mathbf{U}}^{n+1} - \overline{\mathbf{U}}^n = \overline{\Delta\mathbf{U}} - g\tau(H + \overline{h}^{n+1/2})\theta\nabla(\eta^{n+1} - \eta^n) \tag{A1}$$

and

$$\eta^{n+1} - \eta^n = -\tau\nabla\cdot(\alpha\overline{\mathbf{U}}^{n+1} + (1-\alpha)\overline{\mathbf{U}}^n) - \tau(\alpha W^{n+1/2} + (1-\alpha)W^{n-1/2}). \tag{A2}$$



Eliminating $\overline{\mathbf{U}}^{n+1}$ between these two equations, one gets the equation on elevation increment $\Delta\eta = \eta^{n+1} - \eta^n$

$$\Delta\eta - g\tau^2\theta\alpha\nabla\cdot((H+\overline{h}^{n+1/2})\nabla\Delta\eta) = -\tau\nabla\cdot(\alpha\overline{\Delta\mathbf{U}}+\overline{\mathbf{U}}^n) - \tau(\alpha W^{n+1/2}+(1-\alpha)W^{n-1/2})$$

In reality, everything remains similar to the vector-invariant case, and the matrix to be inverted is the same.

– Correct the transport velocities as

$$\mathbf{U}^{n+1} - \mathbf{U}^n = \Delta\mathbf{U} - g\tau h^{n+1/2}\theta\nabla\Delta\eta. \tag{A3}$$

– Proceed with ALE and determine $w^{n+1}$, $h^{n+3/2}$, $T^{n+3/2}$.

– The new velocities are estimated as

$$\mathbf{u}^{n+1} = \mathbf{U}^{n+1}/h^{n+1}.$$

Here $h^{n+1}$ can be computed either in the agreement with the ALE procedure ($\eta^{n+1}$ is already known) or interpolating between $n+1/2$ and $n+3/2$ time levels.

It should be clear now that the vector invariant form can be treated with $h^* = h^n$, but this will require considering both $\mathbf{u}$ and $\mathbf{U}$.

## Appendix B: Subcycling instead of solver

We discuss modifications needed to solve for the external mode through subcycling. This option will be added in future when needed for massively parallel runs. We use the flux form of momentum advection as an example. We take

$$\eta^n = (\overline{h}^{n-1/2} + \overline{h}^{n+1/2})/2,$$

since it provides the second-order accurate estimate.

We follow a common technology and run subcycles between time levels $n$ and $n+2$, with subsequent averaging to level $n+1$. We formally take $\theta = 1$ in vertically averaged equations, for the accuracy of external time stepping will be defined by the procedure used for subcycling. Furthermore, $\eta^{n+1}$ will not be used, but the barotropic part of the new velocity will be directly adjusted.

For the same reason, the contribution from the elevation $\eta^n$ can be omitted while predicting $\Delta\tilde{\mathbf{U}}$. However, if this is done, the implicit solve for vertical viscosity has to be moved to the end, and applied to trim the full velocity $\mathbf{u}^{n+1}$. We will keep the contribution from $\eta^n$ in the predictor step. Then the compensation term with $\eta^n$ will be present (see (B2) below).

Instead of (A1) and (A2) we introduce subcycles indexed with $j$, $j = 0 : 2J$, with $\eta^{n+j/J}$ shortcut to $\eta^j$ and same for $\overline{\mathbf{U}}$ in several formulas below. The simplest form of subcycling looks like

$$\eta^{j+1} - \eta^j = -(\nabla\cdot\overline{\mathbf{U}}^j + W^j)\tau/J. \tag{B1}$$





$$\overline{\mathbf{U}}^{j+1} - \overline{\mathbf{U}}^{j} = \overline{\Delta\mathbf{U}}/J - g(\tau/J)(H + \overline{h}^{n+1/2})\nabla(\eta^{j+1} - \eta^{j}). \tag{B2}$$

Other forms of subcycling can be used to increase stability and reduce the number of subcycles $2J + 1$. The contribution from the Coriolis acceleration can be put in the subcycling procedure (it is zero-order term defining the properties of surface inertia-gravity (Poincaré) waves). To do this we have to (i) move the implicit viscosity update to the end of velocity step, (ii) separate the Coriolis contribution in $\Delta\mathbf{U} = (\Delta\tilde{\mathbf{U}} + f\mathbf{k}\times\mathbf{U}^{AB}) - f\mathbf{k}\times\mathbf{U}^{AB}$, and use the vertically integrated combination in the brackets in place of $\overline{\Delta\mathbf{U}}$ above. If we take the Coriolis acceleration in the barotropic equations, we can also treat it implicitly for better stability.

On completing sybcycles one is at time level $n+2$. In order to eliminate possible high frequencies, averaging is done to time level $n + 1$:

$$\overline{\mathbf{U}}^{n+1} = (2J + 1)^{-1}\sum_{j}\overline{\mathbf{U}}^{j}, \quad \eta^{n+1} = (2J + 1)^{-1}\sum_{j}\eta^{j}.$$

The common further action is to use $\overline{\mathbf{U}}^{n+1}$ for the barotropic transport combined with the baroclinic transport diagnosed from $\mathbf{U}^{n+1}$. We introduce first the new baroclinic transport by writing

$$\mathbf{U}^{*} = \mathbf{U}^{n} + \Delta\mathbf{U},$$

$$\tilde{\mathbf{U}}^{n+1} = \mathbf{U}^{*} - \overline{\mathbf{U}}^{*}\frac{h^{n+1}}{H + \eta^{n+1}}.$$

It is then updated to the full transport velocity by

$$\mathbf{U}^{n+1} = \tilde{\mathbf{U}}^{n+1} + \overline{\mathbf{U}}^{n+1}\frac{h^{n+1}}{H + \eta^{n+1}}.$$

As an aside, we document another possibility which implements a pseudotime solver. We want to solve the same pair of equations as (A1) and (A2). We rewrite these equations as an iterative procedure, with $\delta$ some large parameter

$$\delta(\overline{\mathbf{U}}^{j+1} - \overline{\mathbf{U}}^{j}) = \overline{\mathbf{U}}^{n} - \overline{\mathbf{U}}^{j} + \overline{\Delta\mathbf{U}} - g\tau(H + \overline{h}^{n+1/2})\theta\nabla(\eta^{j} - \eta^{n}),$$

$$\delta(\eta^{j+1} - \eta^{j}) = \eta^{n} - \eta^{j} - \tau\nabla\cdot(\alpha\overline{\mathbf{U}}^{j+1} + (1 - \alpha)\overline{\mathbf{U}}^{n}).$$

In this case $j$ becames a 'pseudotime' index, while the lhs in each of the equations is the residual of iterative process. The analysis of stability shows that one should select $\delta^{2} > k^{2}\tau^{2}c^{2}\theta\alpha$. Here $c$ is the phase speed and $-k^{2}$ is the eigenvalue of the Laplacian operator. Its maximum value is $(\pi/\Delta x)^{2}$. Clearly, damping of fast waves in pseudotime follows $e^{-j/\delta}$, which means that the number of pseudotime iterations $J$ should exceed $\delta$. The hope is that $J$ needs not to be too large if the procedure is kept stable through appropriate selection of $\delta$. The high-frequency waves will be damped over several time steps. The condition on $\delta$ is that it is larger than the Courant number $k\tau c$ (which is much larger than one for $\tau$ of 'internal' mode).





While this option is not cheaper than the commonly used one, it is equivalent to the solution based on semi-implicit solvers and warrants consistency. Indeed, in this case $\Delta\tilde{\mathbf{U}}$ appears as an auxiliary variable, and the issue of barotropic – baroclinic splitting is not emerging.

**Appendix C: Terrain following meshes**

Meshes combining $z$- and terrain-following layers are of interest for studies focused on exchanges between ice cavities or ocean shelves with the deep ocean, and may lead to an improved representation of overflows. The use of tetrahedral elements in previous versions of FESOM was dictated by the need to maintain this functionality. In the framework of ALE this possibility is realized by

prescribing the initial thicknesses of layers as $h_k = h_k(x,y)$ in such a way that some of them follow topography. The practical question is on time step limitations and suppression of dynamical biases on such meshes. We need (i) to adjust the algorithm of computing pressure gradient and (ii) to implement stable isoneutral biharmonic diffusion operators, as suggested by Lemarié et al. (2012a, b). The former means that $\nabla p/\rho_0 + g\rho\nabla Z/\rho_0$ in dynamical equations, which is $\nabla_z p/\rho_0$, may turn

to be insufficiently accurate if discretized as written. FESOM1.4 does not use this two-term representation, but applies vertical polynomial interpolation to the density field instead. This approach will be retained in FESOM2. The implementation of (ii) will allow us to avoid excessive mixing accompanying advection on terrain following meshes. These measures are the subject of ongoing work.

**Appendix D: Isoneutral diffusion on triangular prisms**

For completeness, we write down the expressions for the horizontal and vertical components of fluxes:

$$\mathbf{F}_h(T) = -K_i(\nabla T + \mathbf{s}\partial_z T),$$

$$F_z(T) = -K_i(\mathbf{s}\nabla T + s^2\partial_z T) - K_d\partial_z T.$$

The terms including $K_i$ are referred to as the isoneutral flux, the remaining term with $K_d$ is the dianeutral flux. To complete the description, the slope has to be expressed in terms of thermal expansion and saline contraction coefficients $\alpha$ and $\beta$,

$$\mathbf{s} = -\frac{-\alpha\nabla T + \beta\nabla S}{-\alpha\partial_z T + \beta\partial_z S}.$$

(Note that $\alpha$ here has other meaning as in the rest of paper.) The discretized isoneutral part of the flux operator $K\nabla_3$ should be zero when applied to the density. The implementation difficulty stems from the fact that the tracers together with $\alpha$ and $\beta$ are located at mid-layers, the vertical derivatives are located at the level surfaces, and the horizontal derivatives are at mid-layers, but at cells

instead of vertices. The estimate of slope at a single point is impossible without extra interpolation,





which will break full consistency. The solution involves triads (see, e.g., Griffies (2004) and Lemarié et al. (2012a)) and variational formulation. Note, however, that the implicit time stepping of the contribution with $s^2 K_i$ in the vertical flux, needed for stability reasons (Lemarié et al. (2012a)), will introduce some errors even in this case.

First, we split each triangular prism of our mesh into subvolumes characterized by unique values of the expansion/contraction coefficients, vertical gradients and horizontal gradients, to form triplets. We obtain 6 subprisms per prism, formed by sections along midplane and by vertical planes passing through centroids and mid-edges.

Next, one writes the dissipation functional. We will use different, but equivalent formulation. Consider the bilinear form

$$6\mathcal{F}(\tilde{T},T) = -\sum_{k,c}\sum_{p=1}^{p=6} A_c h_{kc}(\nabla_3\tilde{T}\mathbf{K}\nabla_3 T)_{kcp}.$$

Here the first summation is over mesh prisms (cells and layers), and the second one, over the sub-
prisms $p$. The volume of each subprism is 1/6 of the volume of the full prism (hence the factor 6 on the lhs). Clearly, $\mathcal{F}(T,T)$ corresponds to total variance dissipation. If $T$ is the isoneutral density and its gradients are expressed in terms of $\alpha$ and $\beta$ as for the slope above, $\mathcal{F}$ vanishes.

The last step is to compute the contribution to the rhs of scalar equation from the diffusion term

$$(R_T)_{kv} = (1/A_{kv})\partial\mathcal{F}/\partial\tilde{T}_{kv}.$$

Here we took into account that we deal with layer-integrated equations, hence the division over the area of scalar cell $v$ instead of division by volume. Writing down the expression for $R_T$ is a rather
tedious task. The result can be reformulated in terms of the discrete divergence of discrete flux. Indeed, $(R_T)_{kv}A_{kv}$ is the volume-integrated rhs, i. e., the sum of fluxes through the faces.

Note that since $\mathcal{F}$ is a bilinear form, the definition of the rhs is always globally consistent. Indeed, the total variance dissipation is $\sum_{k,v} T_{kv}(R_T)_{kv}A_{kv} = \sum_{k,v} T_{kv}\partial\mathcal{F}/\partial\tilde{T}_{kv} = \mathcal{F}(T,T)$.

In summary, the variational formulation originally proposed for quadrilaterals can easily be ex-
tended to triangular meshes. All symmetry properties will be granted if computations are local on subprisms.

Substituting $\mathbf{K}$ in the form $\mathcal{F}$ we get

$$\mathcal{F} = \sum_{k,c}\sum_{p}[-K_I\nabla\tilde{T}\cdot\nabla T - K_i\nabla\tilde{T}\cdot\mathbf{s}\partial_z T - K_i\partial_z\tilde{T}\mathbf{s}\cdot\nabla T - (K_d + s^2 K_i)\partial_z\tilde{T}\partial_z T]_{kcp}(A_c h_{kc}/6).$$

The first term does not involve the slope and will not be considered.

Let us start from the third term and compute its contribution to $\partial\mathcal{F}/\partial\tilde{T}_{kv}$. The vertical derivative at level $k$ (the top surface of layer $k$) is

$$(\partial_z T)_{kv} = \frac{T_{(k-1)v} - T_{kv}}{Z_{(k-1)v} - Z_{kv}},$$



and $\nabla T$ is defined on cell $c$

$$(\nabla T)_{kc} = \sum_{v(c)} \mathbf{G}_{cv} T_{kv},$$

Hence it follows for the contribution from layer $k$ and element $c$

$$\frac{\partial \mathcal{F}}{\partial \tilde{T}_{kv}} : \quad \frac{1}{6} A_c h_{kc} \left[ \frac{-1}{Z_{k-1} - Z_k} (-K_i \mathbf{s})_{kcv}^t (\nabla T)_{kc} + \frac{1}{Z_k - Z_{k+1}} (-K_i \mathbf{s})_{kcv}^b \cdot (\nabla T)_{kc} \right],$$

$$\frac{\partial \mathcal{F}}{\partial \tilde{T}_{(k-1)v}} : \quad \frac{1}{6} A_c h_{kc} \frac{1}{Z_{k-1} - Z_k} (-K_i \mathbf{s})_{kcv}^t \cdot (\nabla T)_{kc},$$

$$\frac{\partial \mathcal{F}}{\partial \tilde{T}_{(k+1)v}} : \quad \frac{1}{6} A_c h_{kc} \frac{-1}{Z_k - Z_{k+1}} (-K_i \mathbf{s})_{kcv}^b \cdot (\nabla T)_{kc}.$$

In the expressions above, indices $k$ and $c$ identify the triangular prism, and the index of vertex $v$ together with the upper index $t$ or $b$ identify the subprism (related to $v$ and either top or bottom of the full prism). The expression $(K_i \mathbf{s})_{kcv}^t$ means that $K_i$ is estimated on level $k$ and vertex $v$, and the slope involves the triplet with $\alpha, \beta$ at $kv$, the vertical derivatives at $kv$ and the horizontal derivatives at $kc$. For $(K_i \mathbf{s})_{kcv}^b$, the pairs of indices are $(k+1)v$, $kv$, $(k+1)v$ and $kc$ respectively.

Now, we combine the contributions from the column associated with cell $c$ that enter the rhs of equation on $T_{kv}$ (they come from prisms $(k-1)c$, $kc$ and $(k+1)c$)

$$\frac{\partial \mathcal{F}}{\partial \tilde{T}_{kv}} : \quad \frac{A_c}{6} \left[ \frac{h_{kc}}{Z_{k-1} - Z_k} (K_i \mathbf{s} \cdot \nabla T)_{kcv}^t + \frac{h_{(k-1)c}}{Z_{k-1} - Z_k} (K_i \mathbf{s} \cdot \nabla T)_{(k-1)cv}^b \right.$$

$$\left. - \frac{h_{kc}}{Z_k - Z_{k+1}} (K_i \mathbf{s} \cdot \nabla T)_{kcv}^b - \frac{h_{(k+1)c}}{Z_k - Z_{k+1}} (K_i \mathbf{s} \cdot \nabla T)_{(k+1)cv}^t \right].$$

We easily recognize here the fluxes through the upper and lower surfaces of scalar prism $kv$ coming from the part shared with prism $kc$. They are thickness-weighed over the cells on both sides. Indeed, $2(Z_{k-1} - Z_k) = h_{kc} + h_{(k-1)c}$ for the top surface and similarly for the bottom.

We continue with the contribution from $-s^2 K_i \partial_z \tilde{T} \partial_z T$. The contribution to equation at $(kv)$ from prisms $(k-1)c$, $kc$ and $(k+1)c$ may come from the following terms in $\mathcal{F}$

$$\frac{A_c}{6} \left[ (-s^2 K_i)_{kcv}^t \frac{\tilde{T}_{(k-1)v} - \tilde{T}_{kv}}{Z_{k-1} - Z_k} \frac{T_{(k-1)v} - T_{kv}}{Z_{k-1} - Z_k} h_{kc} + \right.$$

$$(-s^2 K_i)_{kcv}^b \frac{\tilde{T}_{kv} - \tilde{T}_{(k+1)v}}{Z_k - Z_{k+1}} \frac{T_{kv} - T_{(k+1)v}}{Z_k - Z_{k+1}} h_{kc} +$$

$$(-s^2 K_i)_{(k-1)cv}^b \frac{\tilde{T}_{(k-1)v} - \tilde{T}_{kv}}{Z_{k-1} - Z_k} \frac{T_{(k-1)v} - T_{kv}}{Z_{k-1} - Z_k} h_{(k-1)c} +$$

$$\left. (-s^2 K_i)_{(k+1)cv}^t \frac{\tilde{T}_{kv} - \tilde{T}_{(k+1)v}}{Z_k - Z_{k+1}} \frac{T_{kv} - T_{(k+1)v}}{Z_k - Z_{k+1}} h_{(k+1)c} \right].$$

Now, performing differentiation with respect to $T_{kv}$, we find

$$\frac{\partial \mathcal{F}}{\partial \tilde{t}_{kv}} = \frac{A_c}{6} \left[ \left( \frac{h_{kc}}{Z_{k-1} - Z_k} (s^2 K_i))_{kcv}^t + \frac{h_{(k-1)c}}{Z_{k-1} - Z_k} (s^2 K_i))_{(k-1)cv}^b \right) \frac{T_{k-1} - T_k}{Z_{k-1} - Z_k} \right.$$



$$+ \left( -\frac{h_{kc}}{Z_k - Z_{k+1}}(s^2 K_i))_{kcv}^b - \frac{h_{(k+1)c}}{Z_k - Z_{k+1}}(s^2 K_i))_{(k+1)cv}^t \right) \frac{T_k - T_{k+1}}{Z_k - Z_{k+1}} \Big].$$

The result is the standard scheme for the vertical diffusion, but the estimates of $s^2 K_i$ are thickness-weighted over contributing layers. The fluxes through the top and bottom surfaces can conveniently be assembled in a cycle over cells and layers.

We return to the horizontal part in the expression for $\mathcal{F}$. Layer $k$ and cell $c$ contribute to $\mathcal{F}$ as

$$\frac{A_c}{6} h_{kc} (\sum_{v(c)} \mathbf{G}_{cv} \tilde{T}_{kv}) \cdot \left[ \sum_{v(c)} \frac{T_{(k-1)v} - T_{kv}}{Z_{k-1} - Z_k} (-K_i \mathbf{s})_{kcv}^t + \right.$$

$$\left. \sum_{v(c)} \frac{T_{kv} - T_{(k+1)v}}{Z_k - Z_{k+1}} (-K_i \mathbf{s})_{kcv}^b \right].$$

For the contribution into equation $kv$ from $\partial \mathcal{F}/\partial \tilde{T}_{kv}$ it is straightforward to prove that it corresponds to the flux of the quantity in the square brackets through the segments bounding the control volume around $v$ inside triangle $c$. Indeed, for geometrical reasons $\mathbf{G}_{cv}$ is $\mathbf{n}_{cv}/h_{cv}$ with $\mathbf{n}_{cv}$ the normal to the edge of $c$ opposing vertex $v$ directed from this vertex (outer for $c$) and $h_{cv}$ the height in $c$ drawn from $v$. This implies that $A_c \mathbf{G}_{cv} = \mathbf{n}_{cv} l_{cv}/2$, where $l_{cv}$ is the length of the opposing edge. Obviously, for the two segments bounding the control volume $v$ inside cell $c$ the sum of normal vectors multiplied with the lengths of segments is $\mathbf{n}_{cv} l_{cv}/2$. Thus, we arrive at flux representation.



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
