# Peer review of "The Finite-volumE Sea ice-Ocean Model (FESOM2)"

_Geoscientific Model Development, 2016_

## Referee Comment (RC1) · Y. J. Zhang (Referee) · 18 Nov 2016

General comments

This paper is essentially a detailed user manual for the new version of global ocean-ice model FESOM2, now based on a Finite-Volume method. The paper is generally well written and the level of detail is adequate for average users (although a bit hard to read in a few places). I fully concur with the main conclusion that 'FESOM2.0 provides a major step forward in establishing unstructured-mesh models as valuable tools in climate research'. I also appreciate authors' honest account of aspects of model details, and therefore think it'd be a great contribution to GMDD.

Specific comments

I give some minor comments and suggestions below 1. Near line 35: I'd think the change from tetraheda to triangular prisms and also new ALE requires some new learning from users? 2. It'd be beneficial to users if the authors clearly list out the main differences from other models, especially those using similar gridding strategy (line 65); 3. What is used to solve Eq. (11)? It'd be nice to number all equations for easy referencing.

Technical corrections

Editorial corrections: 'looking'->'look' (line 120); what's '()' near line 190? '??' near line 350 needs to be specified. 'described'->'description' (line 440). 'surface'->'subsurface' (line 610).

---

## Referee Comment (RC2) · S.C. Kramer (Referee) · 28 Nov 2016

General Comments:

The main contribution from this paper is the presentation of FESOM 2.0 which is argued to provide a major step forward in establishing unstructured-mesh models as valuable tools in climate research. The previous, finite element version of FESOM has already proven itself as a well established global ocean model and provided evidence for the viability of unstructured mesh methods in this field, with algorithms that are efficient enough to make long-term time-integration feasible. Since this paper shows that the finite-volume FESOM 2.0 model brings a further big improvement in efficiency, I fully agree with this main conclusion and regard the paper to bring a significant scientific contribution.

The paper is generally well written, but some of the numerical details are a little hard to follow. This is to some extent to be expected, as a fully featured, complex model as FESOM brings together a whole range of different methods and techniques. There is an honest discussion of the advantages and drawbacks of the methods that were chosen. I would however have preferred some more focus on the verification and validation of the model. There are many statements on the theoretical properties of the schemes, such as conservation properties, the second order accuracy of the scalar equations, etc., but the paper does not provide benchmarks that test these properties individually. Such tests are important to show a correct implementation, and also that the underlying assumptions of the theory are valid in the relevant regime. Although many details on the numerical discretisation are provided, other important model implementation details are only summarily discussed, for instance the solution strategy for the external mode which, as indicated, has a significant impact on the overall performance and scalability of the model. Also a more in-depth parallel scaling analysis would be of interest.

I realize that at this stage, my recommendations would only add to an already lengthy manuscript and therefore merely suggest to address these in further publications. The main validation of the model, showing that global modelling results of FESOM 2.0 are of at least the same quality as that of FESOM 1.4, but with a significantly increased computational efficiency, is convincing. I only have some very minor comments and corrections that I ask to be addressed.

Specific Comments: - figure 1 is unclear. I think it purports to show both the control volume around a vertex and the vector 'l' directly connecting two cell centres (which doesn't coincide with the control volume edges). It would be better to show these in two separate figures instead. - Although it is understood that the details of the spherical coordinate system are left out, he current paragraph (lines 102-107) is a little hard to parse. Phrases like "The metrics is taken cell-wise constant", "vectors l are stored in radion measure" are just not clear. - this might be obvious, but could you explain why

"In this case the only safe option is to use no-sip boundary conditions" (first paragraph section 2.3, unnumbered for some reason) - the indices indicating column and layer are some times omitted for brevity. However equations (1-2) would become a lot clearer if the k subscripts were included - I do not understand the sentence "However, for this expression..." (line 365) in relation to the previous sentence. - It is claimed (line 415) that Ringler and Randall's ZM discretisation does not ensure momentum conservation. Is that correct? As far as I can see, it is perfectly possible to write down a local discrete momentum balance based on the triangles surrounding the vertices of the hexagon in which the velocities are stored, with fluxes between the triangles that are clearly defined.

Technical comments: Some minor corrections: - page 8, line 211, just before eqn. (4): "layer-intergated" should be "layer-intergrated" - page 7, line 159: "The components of *the* 3D gradient.." - page 19, line 449: "As *a* result..." - page 24, line 565: "at the same time, error*s* become larger ..." - page 12. line 275 garantee -> guarantee - page 19, line 448: "As *a* result" - page 19, line 450: logics -> logic - page 25, line 612: subtropial -> subtropical - page 29, line 711: discretization - line 829: becames -> becomes

---

## Author Comment (AC1) · 8 Dec 2016

article Dear Joseph,

Many thank for your comments on our manuscript. Below we present answers or describe the changes made in the revised version.

1. *Near line 35: I'd think the change from tetraheda to triangular prisms and also new ALE requires some new learning from users?*

   The tetrahedral discretization of FESOM1.4 is hidden from its users, because

the data points are at the prismatic mesh. We agree, of course, that the change in data placement and varying layer thicknesses of ALE vertical coordinate imply some learning, but this issue is largely handled through updating our post-processing scripts in a centralized way. Meshes, forcing, the organization of input and output are inherited from FESOM, and we tried to keep consistency with respect to namelists.

The text of manuscript is adjusted as follows: "It works on the same general triangular meshes and is conceived so as to minimize new learning required from users having experience with FESOM1.4."

2. *It'd be beneficial to users if the authors clearly list out the main differences from other models, especially those using similar gridding strategy (line 65).*

The text is changed to: "The same cell-vertex placement of variables is also used in FVCOM (Chen et al. (2003)), however FESOM2 differs in almost every numerical aspect, including the implementation of time stepping, scalar and momentum advection and dissipation (see below)."

3. *What is used to solve Eq. (11)?*

Equation (11) (old numbering) is solved by using pARMS, as detailed section 5.5. The operator matrix is updated on each time step. The preconditioner is not updated, which works well for global applications.

4. *It'd be nice to number all equations for easy referencing.*

We originally numbered only the equations that are referenced. We follow reviewer's recommendation in the revision, however, we still do not number auxiliary equations.

5. Technical corrections – Done.

With best regards,

The authors

---

## Author Comment (AC2) · 8 Dec 2016

article  Dear Stephan,

We are indebted very much for your comments and suggestions.  Below we present answers and describe changes made in the revised manuscript.

*General comment:*
*I would however have preferred some more focus on the verification and validation of the model. There are many statements on the theoretical properties of the schemes, such as conservation properties, the second order accuracy of the scalar equations,*

[Figure]

*etc., but the paper does not provide benchmarks that test these properties individually. Such tests are important to show a correct implementation, and also that the underlying assumptions of the theory are valid in the relevant regime. Although many details on the numerical discretisation are provided, other important model implementation details are only summarily discussed, for instance the solution strategy for the external mode which, as indicated, has a significant impact on the overall performance and scalability of the model. Also a more in-depth parallel scaling analysis would be of interest.*

Answer:

Many tests and validations were in fact carried out, but we are indeed not in position to include them into the manuscript without doubling its size. We therefore only illustrate the main point that the model is far beyond the dynamical-core phase, it is already a fully functioning model of global ocean circulation.

We of course checked the conservation of volume and tracers, this is an easy task for any errors are immediately seen in balances. The convergence order of scalar transport equations and errors associated with particular algorithms is a topic on its own, and we continue to work on new algorithms. We carried out some elementary tests of many algorithms using shearing velocity fields, and can share our results on request. Most of them show second-order scaling, as expected for varying velocity field. In addition to the issues mentioned in the general comment, there are many others related to the sensitivity to parameterizations, boundary conditions, topography representation in realistic configurations; they are the subject of ongoing work. We also work now on extensive comparisons with FESOM1.4, and on quantifying the differences in the global circulation created by using different options in FESOM2. It is planned to address all this in future papers.

We also agree that some places are presented only schematically, which was necessary to keep paper size limited. To solve for external mode we use pARMS, the operator

matrix is reassembled on each time step, but preconditioner is done only once, to minimize the costs. This works well in configurations relevant for global ocean simulations.

The in-depth parallel scaling analysis is also a topic we continue to work on. For FESOM1.4 we generally observe nearly linear scaling down to 300-400 surface vertices per core. We hope to have similar behavior in FESOM2, but we haven't tested it yet on really large meshes (5-10 M surface nodes, which corresponds to global 1/10 -1/12 degree quasi-Mercator meshes), although we run such meshes with FESOM1.4. Additionally, on finer horizontal meshes one will also use more vertical layers, which may require some adjustments in the strategy of mesh partitioning. Finally, on finer meshes the convergence of ice EVP solver (used by us) would require more subcycles per time step, making the ice model progressively more expensive (with the cost comparable to that of the ocean) and increasing the number of communications needed per time step. This may raise additional questions. We keep all them in mind, and will share our experience in due time. In the manuscript our intension was only to illustrate that the performance we have now is already sufficiently good.

*Specific comments:*

1. *Figure 1 is unclear. I think it purports to show both the control volume around a vertex and the vector 'l' directly connecting two cell centres (which doesn't coincide with the control volume edges). It would be better to show these in two separate figures instead.*

   We edited this figure, splitting it in two panels as suggested, and hope that is serves its purpose better now.

2. *Although it is understood that the details of the spherical coordinate system are left out, the current paragraph (lines 102-107) is a little hard to parse. Phrases like*

*"The metrics is taken cell-wise constant", "vectors l are stored in radian measure" are just not clear. - this might be obvious, but could you explain why.*

We use a local Cartesian reference frame on each triangle, and use cosine of latitude estimated at the triangle center. This has been meant under the cell-wise constant metric, and it is sufficient for the low-order discretization we are using. Scalar gradients are computed on triangles using their local frames, and so are the vectors $d_{ec}$ (each one is computed on $c$ it is related to). Cell areas are computed in the local frame, and scalar areas are computed as sum of respective fragments of cell areas. It can be shown that this ensures consistency between the gradient and divergence operator (they are negative adjoint of each other in energy norm), and also ensures that curl of gradient is identically zero. In reality, the rule here is the same as in most other codes: the metric coefficients have to be estimated at the locations of transport velocity.

Since vectors $l_e$ associated to edges separate two triangles with different cosines, it is more convenient to store their $dx, dy$ in their original radian measure (we use spherical coordinates). In circumstances when their physical lengths are needed, they are computed by using cosines averaged over two triangles sharing the edge.

We added brief explanation to the text.

3. "In this case the only safe option is to use no-slip boundary conditions" (first paragraph section 2.3, unnumbered for some reason)

   It is seemingly a problem of the provided document class.

4. *the indices indicating column and layer are some times omitted for brevity. However equations (1-2) would become a lot clearer if the k subscripts were included*

We added indices $k$ to these equations. We were seeking a compromise between overburdening quantities with indices and the precise sense, but agree that adding the indices helps in this case.

5. *I do not understand the sentence "However, for this expression..." (line 365) in relation to the previous sentence.*

   We meant that in spite of the lack of invariance, using this expression is advantageous, for in this case we can strengthen local connections in the viscous operator.

   The text is modified as: "In spite of this drawback, using the simplified form is advantageous because the contributions from the neighbor velocities in flux divergence can be strengthened. Indeed, only contraction with normal vector ..."

6. *It is claimed (line 415) that Ringler and Randall's ZM discretisation does not ensure momentum conservation. Is that correct? As far as I can see, it is perfectly possible to write down a local discrete momentum balance based on the triangles surrounding the vertices of the hexagon in which the velocities are stored, with fluxes between the triangles that are clearly defined.*

   The ZM discretization on its own has no problems, it is the vector Laplacian defined on the stencil involving (in the language of triangular cell-vertex discretization)) triangle $c$ and its three nearest neighbors $n_1, n_3$ and $n_3$. The point is that the estimate of Laplacian *on such a stencil* is not related to the control volume $c$, but to a smaller control volume. In order to compute the vector Laplacian $\Delta u = \nabla\nabla \cdot u - \nabla \times \nabla \times u$ based on just 4 $u$ values (on triangle $c$ and its three neighbors $n_1, n_3$ and $n_3$) one considers first three triangles formed by connecting the centers of $c, n_1, n_2$, the centers of $c, n_2, n_3$ and the centers of $c, n_3, n_1$. On each of them the divergence and curl can be computed. Then, the vector Laplacian can be estimated at the triangle formed by connecting the centers of these

three triangles. This control volume differs from the control volume of the original cell $c$. This difference does not matter on uniform meshes, where the triangle will be similar to $c$, but it matters on general meshes. Furthermore, any attempt to use varying $\nu_h$ is also incompatible with momentum conservation.

It is of course perfectly possible to write down a local discrete momentum balance based on the triangles surrounding the vertices (for example, viscous operator (12) in old version of the manuscript, satisfies momentum balance), but on general meshes this would not give the Laplacian operator, because four points are not enough for that. The small-stencil Laplacian represents the Laplacian operator, but not on the control volume one really needs.

7. *Technical comments: Some minor corrections:*

We corrected them, many thanks.

With best regards,

The authors